# Binge Eating Disorder Is a Social Justice Issue: A Cross-Sectional Mixed-Methods Study of Binge Eating Disorder Experts’ Opinions

**DOI:** 10.3390/ijerph19106243

**Published:** 2022-05-20

**Authors:** Brenna Bray, Chris Bray, Ryan Bradley, Heather Zwickey

**Affiliations:** 1Helfgott Research Institute, National University of Natural Medicine, Portland, OR 97201, USA; rbradley@nunm.edu (R.B.); hzwickey@nunm.edu (H.Z.); 2Wilder Research Division, Amherst H. Wilder Foundation, Saint Paul, MN 55104, USA; bray0021@umn.edu; 3Herbert Wertheim School of Public Health, University of California, San Diego, CA 97037, USA

**Keywords:** binge eating disorder, binge eating, oppression, diversity inclusion, minority, economic insecurity, stigma, trauma, social media, food scarcity

## Abstract

Background: Binge eating disorder is an autonomous DSM-V diagnosis characterized by discrete rapid consumption of objectively large amounts of food without compensation, associated with loss of control and distress. Environmental factors that contribute to binge eating disorder continue to evolve. This mixed-methods cross-sectional study assessed whether there is consensus among experts in the field about environmental factors that influence adult binge eating disorder pathology. Methods: Fourteen expert binge eating disorder researchers, clinicians, and healthcare administrators were identified internationally based on federal funding, PubMed-indexed publications, active practice in the field, leadership in relevant societies, and/or clinical and popular press distinction. Semi-structured interviews were recorded anonymously and analyzed by ≥2 investigators using reflexive thematic analysis and quantification. Results: Identified themes included: (1) systemic issues and systems of oppression (100%); (2) marginalized and under-represented populations (100%); (3) economic precarity and food/nutrition insecurity/scarcity (93%); (4) stigmatization and its psychological impacts (93%); (5) trauma and adversity (79%); (6) interpersonal factors (64%); (7) social messaging and social media (50%); (8) predatory food industry practices (29%); and (9) research/clinical gaps and directives (100%). Conclusions: Overall, experts call for policy changes around systemic factors that abet binge eating and for greater public education about who can have binge eating disorder. There is also a call to take and account for the narratives and life experiences of individuals with binge eating disorder to better inform our current understanding of the diagnosis and the environmental factors that impact it.

## 1. Introduction

Binge eating disorder is an autonomous DSM-V diagnosis characterized by discrete rapid consumption of objectively large amounts of food without compensation, associated with loss of control and distress [1]. The disorder has high lifetime prevalence rates (5%–13% [2,3]) and a complicated health sequelae [4,5] associated with genetic and environmental factors [6,7]. Several models of environmental factors for binge eating disorder have been proposed [7]. These models focus on behavioral genetic data that consistently support the contributions of both genetic and environmental influences to eating disorders and symptoms. For example, case-control family and twin-based studies have estimated heritability for binge eating disorder to be between 39% and 57% [8,9,10,11], with 42%–59% of variance attributed to unique environmental factors (not shared by co-twins) [9,12] and 13% attributed to common environmental factors [9]. These studies have not had complete agreement on whether shared environment plays a role in binge eating disorder [8]. Moreover, these studies were conducted between 2004–2010 [8,9,10,11] and are now outdated. Sociocultural models that focus on thinness ideals also have empirical support for increasing risk of eating disorder cognitions and behaviors in adolescents and women with eating disorders at large [7,13,14]. However, these models often lack empirical testing in populations of individuals with binge eating disorder, and in men and other minority populations.

More recently, social justice (equal distribution of opportunity, privileges, and wealth within a society [15]) has gained attention and traction socioculturally and in research broadly [16,17] as well as in the field of eating disorders specifically [18,19,20,21]. For example, the Coronavirus 19 (COVID-19) pandemic has caused an increase in observational studies that have helped improve our understanding as a field of the experience of binge eating disorder from those who experience it [18,20] and tend to identify important social justice issues that seem highly relevant to binge eating disorder pathology [18,19,20,21]. These include eating disorder stigmatization and lack of diversity, inclusion, and representation for individuals with marginalized identities [20]. A recent review on binge eating disorder epidemiology [19] noted that less than half of adults with binge eating disorder are recognized in healthcare and minority status, deprivation, violence, trauma, and major mental health illnesses were identified as possible risk factors for binge eating disorder [19]. However, these proposed environmental risk factors lack empirical testing and more information on the contributions of environmental factors contributing to binge eating disorder is warranted and needed. Therefore, we interviewed 14 systematically identified expert binge eating disorder researchers, clinicians, and healthcare administrators on their perspectives of adult binge eating disorder pathology and treatment broadly. Here, we report on environmental factors identified as relevant to adult binge eating disorder pathology through reflexive thematic analysis of the semi-structured interviews that were conducted and recorded anonymously with the 14 systematically identified experts in the field.

## 2. Materials and Methods

### 2.1. Participants and Recruitment

This study recruited researchers, clinicians, and healthcare administrators with expertise in adult binge eating disorder. Eligibility criteria is shown in Table 1.

### 2.2. Procedure

With approval from the National University of Natural Medicine (NUNM) IRB (# HZ12120), Dr. Bray sent eligible participants a scripted email study invitation. Respondents were interviewed anonymously on Zoom (Zoom.com; last accessed 19 May 2022) with verbal consent obtained at the start of each interview. Most interviews were scheduled for two hours, with abbreviated 30–60-min interviews conducted as needed. Table 2 shows interview questions pertaining to binge eating disorder pathology. Demographic information was collected at the end of each interview verbally or through follow-up email survey.

### 2.3. Data Analysis

Interview recordings were transcribed. Transcripts were de-identified and two investigators (BB and HZ) separately reviewed and qualitatively analyzed each transcript for common themes among interviews using a reflexive thematic analysis approach [29]. Interviews were independently coded by BB and HZ. Themes were identified independently then discussed and finalized through reflexive engagement with the data [29] with intentional involvement of BB and HZ. Transcripts were analyzed quantitatively by BB to identify the number of participants who expressed positive/supportive, negative/skeptical, or neutral perspectives on each of the identified themes. Additional investigators were consulted when quantitative analysis questions arose and for tiebreakers (HZ, CB).

### 2.4. Participant Response Rates and Characteristics

Thirty-eight experts met enrollment criteria and fourteen consented, enrolled, and participated in the study (Figure 1). This included six individuals who met the academic/research criteria (6/14, 43%), five who met the clinical criteria (5/14, 36%), one who met both the academic/research and clinical criteria (1/14, 7%), and two who met some criteria from the academic- and clinical categories so as to qualify for inclusion in a mixed option (2/14, 14%) (Table 1). Characteristics for the 13/14 participants who provided their demographic information are provided in Table 3.

## 3. Results

### 3.1. Theme 1: Systematic Issues and Systems of Oppression (14/14, 100%)

All fourteen participants (14/14, 100%) addressed systemic issues related to binge eating disorder (Table 4). Twelve subthemes were identified that addressed singular or intersectional systems of oppression, all of which were described as causative or comorbid with binge eating disorder or impeding research, treatment, or remission (Table 4). The subthemes were: (i) systematic discrimination (e.g., body weight/shape/size discrimination (see Theme 4), systematic racism, systematic sexism) (12/14, 82%); (ii) media messaging and sociocultural mandates (e.g., perpetuating stigmatization, body weight/shape/size ideals and discrimination, “diet culture,” and movement and fitness ideals; see Theme 7; 12/14, 82%); (iii) insurance and healthcare systems (including insurance costs and coverage, treatment costs, systematic stigmatization from healthcare providers, geographic access to treatment, mandated movement for individuals in larger bodies, and provider scarcity; 9/14, 64%); (iv) “predatory” food industries/environments (see Theme 6; 4/14, 29%); (v) abuse (often domestic, including sexual, emotional, and/or physical; Theme 5, 4/14, 29%); (vi) geographical systems (e.g., geographical inequities in provider and treatment access, and in government assistance programs like the Supplemental Nutrition Assistance Program (SNAP) that can limit its effectiveness [30,31,32]; 4/14, 29%); (vii) eating disorder research as a field (e.g., operating from an anorexic-centric perspective/understanding; 3/14, 21%); (viii) research funding (e.g., lack of funds for eating disorder research relative to disorders of similar prevalence, lack of clarity regarding what agencies should fund eating disorder research; 2/14, 14%); (ix) economic exploitation (a relationship in economic wealth distribution wherein a worker does not receive proper compensation for his/her work [33]; 1/14, 7%); (x) school systems (e.g., individuals in larger bodies being less likely to be called on in class or receive college entrance; 1/14, 7%); (xi) legal systems (1/14, 7%); and (xii) police harassment (1/14 7%).

“If you work with eating disorders, it’s a political statement, [especially] as we’re [better] understanding [the impacts of] racial injustice. … targeting inequity, [is] going to have a cascading effect across mental illnesses, and especially the [way] we understand the impact of discrimination. … there’s no question we need to be thinking about …broad-based social factors related to poverty, related to stigma and weight stigma, obviously in binge eating, but frankly, you know, all types of stigma, all types of discrimination, because binge eating … crosses demographics, gender… [and] … my guess is binge eating disorder is horrifically under diagnosed among people who aren’t [currently] on our radar at all.” (P72)

“We are living in a society that’s so weight-focused and oppressive towards individuals living in like larger bodies… [and] …a lot of times, the demand to move has been …an oppressive demand upon the body and the psyche that is founded and shaming the person.” (P7)

### 3.2. Theme 2: Marginalized and Under-Represented Populations (14/14, 100%)

All fourteen participants (14/14, 93%) acknowledged marginalized and minority populations that are disproportionately impacted, screened, treated, and represented in the field (Table 5). Subthemes pertaining to historically marginalized and overlooked populations included: (i) individuals with low socioeconomic status (13/14, 93%; see Theme 3.i below); (ii) individuals with food or nutrition scarcity (10/14, 71%; see Theme 3,ii–iii below); (iii) male sex/gender (8/14, 57%); (iv) racial and ethnic minorities, including black, indigenous, and people of color (5/14, 36%), (v) lesbian, gay, bisexual, transgender, and nonbinary populations (3/14, 21%), (vi) age (2/14, 14%); and (vii) religion (1/14, 7%).

“If you’re a black woman, if you are somebody who lives in a larger body, if you are an older male, people aren’t going to think that your eating habits [constitute] an eating disorder because you aren’t … a young, thin, cis-gendered, white woman, and so I think that even just recognizing that binge eating disorder is ‘a thing’ is one of those things that gets in the way… I think it also makes sense to talk about the specific ways in which underrepresented groups might be struggling that are unique to them and their experiences versus just saying ‘anyone can be affected.’ …So knowing, for example, that if you are a sexual or gender minority you are at much greater risk of any kind of eating disorder behaviors, knowing that if you are a BIPOC member of a community you are very unlikely to get detected with an eating disorder, and that means you could struggle for a long time, and what do we do then to reach these communities in a way that’s meaningful?” (P75)

### 3.3. Theme 3: Economic, Food, and Nutrition Scarcity and Insecurity (13/14, 93%)

#### 3.3.1. Subtheme i: Economic Precarity (93%)

Thirteen participants addressed economic aspects of binge eating disorder (13/14, 93%); five (5/14, 36%) described direct connections between binge eating disorder pathology and economic status/precarity (Table 6). Seven factors were described as potentially mediating or moderating the relationship between economic precarity and binge eating pathology. These included: food insecurity (5/14, 36%); nutritional access/insecurity (5/14, 36%); food environment (3/14 participants, 21%), mental health risks (2/14, 14%), the COVID-19 pandemic (2/14, 14%), access to treatment resources (2/14, 14%), and weight biases (stating individuals in larger bodies experience economic discrimination) (1/14, 7%). Many of the economic factors—including the relationship between financial security, food scarcity/insecurity, and a pattern of forced restriction and binge eating—were endorsed as “systemic” public health issues.

“If [an individual’s] economic status is not as good, their physical health status will be not as good and their mental health status will be not as good and we know that socioeconomic disadvantage is a major player in terms of risk for …perpetuation of illness, for maintenance of illness. And sometimes for, … the onset of illness as well. …socioeconomic disadvantaged groups in the community have poorer mental health, we know that. So, it is really important.” (P93)

#### 3.3.2. Subtheme ii: Food Scarcity & Insecurity (64%)

Nine participants (9/14, 64%) identified food scarcity and insecurity—in childhood or adulthood—as being relevant to binge eating disorder pathology when present (Table 6). Five participants (5/14, 36%) cited research findings linking food insecurity to binge eating disorder (three specifically (e.g., referencing specific author(s) and findings [34,35]) and two broadly (e.g., referenced literature without specific author(s)). Food scarcity and insecurity were described as potentially disrupting one’s relationship with food or eating (5/14, 36%) and increasing risk for other physical and psychological health problems (4/14, 29%). Food scarcity/insecurity were linked to economic insecurity (5/14, 36%) and the COVID-19 pandemic (2/14, 14%). One participant (1/14, 7%) described childhood adverse food experiences as important adverse childhood experiences (ACEs) that are often overlooked and under-screened, but that potentially relate to adult eating disorder pathology.

“I work with patients who have said, ‘well yeah, I have binge eating. I binge eat the first two weeks of the month ‘cause that’s when we have food in the house and then there’s no food in the house the last two weeks of the month.’ That’s a systemic issue that I think needs to be addressed and needs to be talked about in terms of people’s vulnerability to eating disorders.” (P75)

#### 3.3.3. Subtheme iii: Nutritional Scarcity & Undernutrition (43%)

Six participants (6/14, 43%) addressed nutrition scarcity and/or undernutrition, linking it to lower socioeconomic status (4/14, 29%) and food environment (3/14, 21%), with potential to reduce access to “healthy, nutritious foods.,” (Table 6). One participant cited three separate areas of research demonstrating that: (a) malnutrition can occur in individuals with larger bodies [36,37,38,39,40], (b) malnutrition can lead to food preoccupation [41], and (c) maternal malnutrition is linked to offspring obesity (see comments in Table 6 legend regarding literature cited) [42,43]. Another participant cited a study relating urbanization factors—including food and nutrition insecurity and poverty—to increased risk for eating disorders, including binge eating disorder [35].

“We know that poor people have less access to nutritious foods and have a food environment that I think is predatory, to push, you know, highly processed foods …Even our fruits and vegetables, because of soil and farming practices have become less nutritious.” (P19)

### 3.4. Theme 4: Stigmatization and Its Psychological Impacts (13/14, 93%)

#### 3.4.1. Subtheme i: Forms of Stigmatization (93%)

Thirteen participants (13/14, 93%) described five different foci of internally- and externally driven stigmatization relevant to binge eating disorder pathology. These included stigmatizations pertaining to: (a) body weight, shape, and size (12/13, 92%); (b) having an eating disorder diagnosis (5/13, 39%); (c) having a mental health diagnosis (4/13, 31%); (d) having any medical diagnosis (1/13, 8%); and/or (e) perfectionistic food/eating ideals (1/13, 8%) (Table 7).

“People with eating disorders, they struggle with mental health stigma, they struggle with eating disorder stigma, and they struggle with weight stigma. [So those are] three [forms of stigmatization that can] obviously can impact [individuals with binge eating disorder] in very severe ways.” (P93)

“… the stereotypical [judgement is] that someone in a larger body is a failure, is lazy, is all these negative things…” (P37)

“Anytime you’re given the message overt or covert, that something is… wrong with you, like you’re bad, you’re too much, you’re big, you’re repulsive, you’re gluttonous, whatever, … that’s really the message that people internalize through a lot of experiences with healthcare practice, practitioners, families, schools, etc.” (P7)

#### 3.4.2. Subtheme ii: Body Weight/Shape/Size Stigmatization (93%)

Six foci pertaining to body weight/shape/size stigmatization (subtheme 4ia) were identified, including body weight/shape/size stigmatization as: (a) potentially exacerbating binge eating disorder symptoms (11/14, 79%); (b) prevalent among healthcare providers and in the medical system (6/14, 43%); (c) core to binge eating disorder pathology (4/14, 29%); (d) area requiring better understanding of its trajectory and impact (4/14, 29%); (e) a form of trauma or adversity, especially when occurring during childhood or chronically (3/14, 21%); (f) an area requiring better understanding of its trajectory and impact (4/14, 29%); (g) possibly varying by ethnicity (suggesting different ethnicities may have different levels of acceptance around weight that impact distress frequency and treatment seeking) (Table 7).

“The police are to black men as the medical establishment is to black women” (P72)

“I have a patient with binge eating disorder whose doctor told her, ‘you’re fat every day, so you should exercise every day.’ That’s from a health care practitioner. But that’s really …what we’re telling people [as a culture/society] …the practitioner just put [it into] words.” (P7)

#### 3.4.3. Subtheme iii: Body Weight/Shape/Size Discrimination (21%)

Three participants (3/12, 25%) also discussed weight discrimination, described as one of the few remaining socially acceptable and legal forms of discrimination in the U.S., affecting college admission, job security, and healthcare treatment (Table 7).

### 3.5. Theme 5: Trauma and Adversity (11/14, 79%)

Eleven participants (11/14, 92%) described trauma or adversity exposure as relevant to binge eating disorder pathology when present (Table 8 including adverse childhood experiences (ACES), early life adversity (ELA), adverse life experiences (ALAs), and post-traumatic stress disorder (PTSD). Several subthemes were identified, including: (i) relevant forms of trauma/adversity (7/14, 50%); (ii) the relationship between trauma/adversity and binge eating disorder pathology (11/14, 79%); and (iii) critical considerations (3/14, 21%) (Table 8).

#### 3.5.1. Subtheme i: Forms of Trauma/Adversity (50%)

Seven participants (7/14, 50%) identified specific forms of trauma or adversity that may relate to binge eating disorder pathology (Table 8). These included: (a) abuse (sexual, physical, or emotional; 4/14, 29%, two specifically referencing early/childhood abuse, 14%); (b) trauma related to body weight/shape/size stigma (3/14, 21%); (c) the COVID-19 pandemic (3/12, 21%); (d) invalidating/oppressive experiences/environments (2/12, 14%); (e) interpersonal trauma (2/14, 14%); (f) trauma related to movement or physical activity (e.g., being forced to run in gym class and ridiculed by peers) (2/14, 7%); (g) childhood food experiences as ACES with traumatic consequences (1/14, 7%); (h) trauma related to chronic dieting (1/14, 7%); (i) living with irritable bowel syndrome as traumatic; and (j) trauma related to other untreated diagnoses (e.g., ADHD) (1/14, 7%).

“[There’s] lots of research showing that traumatic early life experiences, sexual abuse, but also other forms of abuse, emotional and physical abuse, increased someone’s risk for an eating disorder.” (P93)

“If the eating disorder has been associated with weight gain, then we know for a fact that they’ve been intruded upon by families, doctors, … institutions, and … there’s trauma associated with that. There’s trauma [associated] with …being told day in and day out that what you are is not acceptable or lovable or okay.” (P7)

“… trauma of physical activity … the idea that they don’t want to work out, but it’s really that their middle school teacher was screaming at them when they were trying to do their …whatever… PE class, or they got made fun of.” (P37)

“Trauma is so bad for the brain and what we’re seeing around ‘little t trauma,’ if you are someone [who is] susceptible, and you are teased and bullied, I think there’s a lifelong consequence for a lot of those individuals, and I think that absolutely sets up the trajectory around eating dysregulation, no question about it.” (P72)

#### 3.5.2. Subtheme ii: Relationship between Trauma/Adversity and Binge Eating Disorder Pathology (79%)

Eleven participants (11/14, 79%) expressed views that trauma exposure is relevant to binge eating disorder pathology, with four participants referencing research findings (4/14, 29%) (Table 8 Seven major trends were identified by which participants described trauma/adversity as being related to binge eating disorder pathology: (a) trauma/adversity as increasing risk for binge eating disorder (5/14, 36%; see Table 9 for supportive subthemes); (b) neurobiological impacts of trauma/adversity may prime binge eating disorder (2/14, 14%); (c) binge eating to cope with trauma/adversity or resulting mood regulation disturbances (2/14, 14%); (d) trauma/adversity as exacerbating or triggering binge eating disorder symptoms (2/14, 14%); (e) trauma/adversity as being comorbid with binge eating disorder (1/14, 7%); and (f) additional possible mechanistic pathways by which trauma/adversity may be linked to binge eating disorder (2/14, 14%), including the possibility that trauma/adversity can “make it harder to tolerate the distress that comes with doing treatment,” (1/14, 7%).

“We know that people who are traumatized can have some upsetting of their arousal responses [that is] biological, and that … managing [and] down-regulating [the hyper-arousal response] is an important part of therapy often. …We know that there are biological brain changes [that occur] as a consequence of repeated adverse life experiences or traumatic experiences. We know that post-traumatic stress disorder is a disorder that is a common outcome, but that people can [also] develop other disorders, which are not PTSD, but that occur as a consequence [of repeated traumatic or adverse life experiences]. …As part of that dysregulation of emotions and those experiences, binge eating can [become] a way of modulating those emotions. … it’s a real phenomenon and a real effect, and lots of research [shows] that traumatic early life experiences, sexual abuse, but also other forms of abuse, emotional and physical abuse, increase someone’s risk for an eating disorder.” (P93)

#### 3.5.3. Subtheme iii: Critical Considerations (36%)

Several critical considerations were addressed related to trauma/adversity in binge eating disorder (Table 8). Four participants (4/14, 29%) expressed the importance of addressing trauma and adversity in treatment; one of these participants specifically expressed an importance for identifying *how* to address a trauma/adversity history in treatment (1/14, 14%). Two participants (2/14, 14%) expressed the importance of screening for trauma and adversity. One participant (1/14, 14%) expressed a need for a greater understanding of the relationship between trauma/adversity and binge eating disorder. One participant (1/14, 7%) referenced literature findings of poor-self report in trauma.

“To what degree do we understand any trauma that somebody with binge eating disorder has experienced throughout their life, either singularly or multiple times? And how does that play a role in … their current experience? And [trauma] can be specific …traumatic events, it can be the ongoing impact of chronic stress related to either low level trauma or the trauma of chronic racism or the trauma of chronic weight stigma. And so how do we think about that and where does that fit into… our treatments?” (P60)

### 3.6. Theme 6: Interpersonal Factors (9/14, 64%)

Nine participants (9/14, 64%) expressed views that interpersonal relationships, effectiveness, and deficits play important roles in binge eating disorder pathology (Table 9).

#### 3.6.1. Subtheme i: Ways Interpersonal Factors Can Impact Binge Eating Disorder (50%)

Seven participants (7/14, 50%) addressed five different ways in which interpersonal deficits or negative interpersonal experiences are relevant to binge eating disorder pathology. These included: (a) social sensitivity related to social anxiety/fear/threat perception (5/14, 29%); (b) interpersonal deficits (e.g., poor communication skills or social interaction abilities (3/14, 29%), (c) socialization around food/eating as problematic social activity (3/14, 21%); (d) social anxiety disorder as a relevant comorbidity in binge eating disorder (3/14, 21%); (e) spousal relationships, intimacy, and sexuality (2/14, 14%); and (f) negative social experiences or deficient communication skills directly catalyzing or contributing to binge eating behavior (2/14, 14%).

“Any form of interpersonal deficits or a struggle in terms of sustaining, maintaining good quality relationships in life and having people [to] confide in is an important vulnerability factor for an eating disorder, but also may probably help explain why interpersonal psychological therapy and addressing interpersonal deficits is an effective treatment in controlled trials.” (P93)

#### 3.6.2. Subtheme ii: Ways Binge Eating Disorder Can Impact Interpersonal Factors (36%)

Five participants (5/14, 36%) described five different ways in which aspects of binge eating disorder may contribute to interpersonal deficits. These aspects included: (a) body weight/shape/size stigmatization (2/14, 14%); (b) body weight/shape/size overvaluation (1/14, 7%); (c) social ranking (1/14, 7%); (d) broader social phenomenon (e.g., systemic discrimination and stigmatization (1/14, 7%); and (e) COVID-19 quarantine/isolation (1/14, 7%).

“As a field … we neglect social anxiety disorder because we tend to think it’s just about weight and shape, self-consciousness, I think we under-diagnose this. … we need to be looking specifically at Social Anxiety Disorder and I think based on Janet Treasurer’s work, we’re going to end up seeing that there’s links in …sensitivity to social threat, … the extent to which that’s causal, secondary to the eating disorder … understanding where anxieties sort of intersect and [understanding the] neurocognitive process …especially around threat sensitivity… is going to be really helpful.” (P72)

#### 3.6.3. Subtheme iii: Positive Relationships between Social Interaction and Binge Eating Disorder Pathology (21%)

Three participants (3/14, 21%) also described positive relationships between social interaction and binge eating disorder pathology (outside of social media and social messaging). Two of these participants (2/14, 14%) referenced the positive impacts of community and one referenced the benefits of family (1/14, 7%). One participant expressed in a neutral, nonspecific manner, that social factors are important in binge eating disorder (1/14, 7%).

“[Social support] has an enormous impact not just on your behavior, but on you know, your brain functioning, honestly, I mean, it means you are in a community you are being cared for you are accountable.” (P72)

### 3.7. Theme 7: Social Messaging and Social Media (7/14, 50%)

Seven participants (7/14, 50%) described social messages (3/14, 21%) and social media (5/14, 15%) as being significantly relevant to binge eating disorder pathology (Table 10). Four participants (4/14, 28%) described the relationship as exclusively negative, primarily by reinforcing ideals around body weight/shape/size, food, eating, and fitness that contribute to stigmatization, social ranking, social interactions, and self-esteem/valuation/negative affect. Three participants (3/14, 21%) described the relationship as primarily negative, but also having some positive aspects or potential.

“If you look at social media, the amount of blaming and stigmatizing and the link still… between … character and weight and shape and the role of thin privilege. I really do believe that if we can shift some of that it’s going to have broader based implications around eating disorders, especially—frankly—binge eating, because people tend to be higher weighted.” (P72)

“Social media is horrible with the way it advertises all kinds of stuff …food advertising and weight stigma, and then diet pills and weight loss products… …people are being bombarded with all of that all the time when they’re online.” (P16)

“You just have to look at social media to realize other people can be part problem as well, as part of the solution.” (P84)

### 3.8. Theme 8: Predatory Food Industry Practices and Environments (4/14, 29%)

Four participants (4/14, 29%) described “predatory” food industry practices, such as hiring engineers to design foods that produce specific rewarding or emotional responses and promote consumption, potentially leading to over-consumption, and binge eating (Table 11).

“The question of political utility is something which we don’t usually talk about in science, but I saw the nutritional epidemiology field paying attention to the emotional aspects of overeating and the emotional and physiological aspects of the way processed foods are created to promote overeating by tapping into physiological responses to fat, sugar, crunch, salt……There’s so many processed foods that are designed to get people to overeat or to … trigger an emotional response that then [makes] someone prone to binge eating as a way of emotionally coping with things that are happening around them that feel out of their control or that are damaging to them. …There’s so many different systems; the food system is one of them.…The food industry and generation of processed foods [are] part of … the landscape that aids and abets binge eating and binge eating disorder. [They impact] what the food landscape looks like and there’s people profiting off of that. During this past 42 min …there are companies and individuals profiting off of making food that will lead someone down a path to binge eating disorder or binge eating… If that’s not a system of oppression I don’t know what is.” (P16)

This was often identified as an issue of political utility and public policy. Comparisons were made between “big food” and “big tobacco.” One participant noted that:

“With … tobacco, …we were working on these treatments, and pharmacology, and all these sorts of things, and we really didn’t start to see drops in [tobacco use] until we changed the tobacco environment. … Leaning on a public health perspective … what have we done … to reduce people’s tendency to overuse [things like tobacco] or with alcohol too … we really focused on altering the environment, so, there’s not as much marketing, there’s not as many triggers, it’s not in your face, it’s not in vending machines, it’s not targeted to kids, it’s more expensive. And so, I think of all of those … environmental interventions… …if you can have a more optimal environment that encourages …healthy eating, and there’s not as much temptation, that allows our individual treatments to have a better chance for success. Because if you’re trying to use … individual treatments to combat a truly oppressive food environment, that’s a really tall order.” (P19)

One participant also advocated for the importance of informing individuals with binge eating disorder of the nature of these foods/food industries and “what they’re up against” to alleviate the sense of failure and guilt that often accompanies binge eating these foods.

“To ignore the fact that the food environment has changed, and that we are all kind of dealing with …predatory industry practices, but with very hyper-engineered, highly rewarding foods, to not acknowledge that in any way is problematic, and I think not giving people …the full picture of what they are dealing with. … [If you’re] sitting down with a bowl of ice cream and a bowl of salad, you’re dealing with fundamentally …very different things from …a reward, and even a psychological … profile. And so, it’s not just that… ‘if you just tried hard enough, … you’d … just be able to sort this out very easily.’ …under-selling the challenge of our food environment and the foods that we’re dealing with, especially [for those who] are under-resourced, [is] not giving people the full picture.” (P19)

One participant (1/14, 7%) acknowledged certain processed/engineered foods (e.g., brownies) can produce different reward- and behavioral responses than natural whole foods (e.g., broccoli), but did not describe these foods as being intentionally engineered to create specific consumer responses, or as contributing to the phenomenon of binge eating. One participant (1/14, 7%) acknowledged a variety of foods can be highly rewarding to a variety of consumers but did not identify these foods as being “processed” or intentionally “engineered.” One additional participant (1/14, 7%) recognized disordered eating behavior can be associated with specific foods but attributed this to the internalization of specific “food rules,” stating that normalizing mindful food consumption can extinguish “food rules” and the disordered eating behavior in turn.

“I have people [who] think that they’re addicted to food. Once we normalize that food, though, then it’s like, ‘oh, okay, I can have cheesecake for my snack. Awesome,’.” (P37)

### 3.9. Theme 9: Research Gaps and Future Directives (14/14, 100%)

Six subthemes were identified regarding gaps in the literature the experts would like to see closed and future research directives that pertain to the themes identified herein (Table 12).

#### 3.9.1. Subtheme i: Systemic Changes (71%)

Ten participants identified a need for change in some of the systems that currently abet binge eating disorder (10/14, 71%). Spontaneously identified systems included: **(a)** the eating disorder field (5/14, 36%), including a need for increased funding for eating disorder research (equally proportionate to that available for research on other disorders of similar magnitude) and need for clarification on which funding agencies should fund eating disorder research (2/14, 14%), change in mandated movement perpetuated by the medical field that can be traumatic for individuals with binge eating (e.g., recommendations and mandates for movement—from PE instructors and medical doctors—given in ways that are insensitive to weight stigmatization and bullying, see Theme 4; 2/14, 14%), and recognition of implicit biases, stigmatizations, and discriminations toward individuals with binge eating and in larger bodies held among those in the field (1/14, 7%). **(b)** Food systems and availability (4/14, 29%), including food industry practices (2/14, 14%) and food stamp allotment (1/14, 7%). **(c)** Other systems of oppression (e.g., structural racism and sexism, economic exploitation (see statements from P16 in Theme 1), and “broader sociocultural issues,” (P72)); and **(d)** economic aspects that prevent treatment access (1/14, 7%).

“I think we’re very underfunded in terms of treatment trials, and …woefully underfunded when we compare ourselves with high weight disorders. … …and just generally, across the board, we need …more funding for research…” (P93)

“I think that there’s a professional socio-cultural administrative framework that is having trouble getting its arms around binge eating disorder…” (P33)

“I think I’ll just go back to the food systems issue, the manufacturers of processed foods. …That’s the one that … we need whole cohorts of graduates from public health schools and psychology and some other fields just to document what’s happening there and work with policymakers to change what food manufacturers are allowed to do.” (P16)

#### 3.9.2. Subtheme ii: Understanding Environmental Impacts (36%)

Five participants addressed a need to understand the role of environmental factors that impact or increase risk for binge eating disorder (5/14, 36%), including the impacts of: (a) mandated movement (2/14, 15%); (b) trauma (1/14, 7%); (c) “broader sociocultural issues” (1/14, 7%) (d) community (1/14, 7%); (e) interpersonal threat/threat sensitivity (1/14, 7%); and (f) environmental pollution (1/14, 7%).

#### 3.9.3. Subtheme iii: Including Marginalized Populations (29%)

Four participants identified a need to include minority and marginalized populations historically overlooked in the field (research and clinically) and societally (4/14, 29%), specifically: (a) including and reaching men (1/14, 7%); (b) including individuals in normal-sized bodies (1/14, 7%); (c) identifying struggles unique to marginalized populations (1/14, 7%); and d) information dissemination (1/14, 7%).

“I think it …makes sense to talk about the specific ways in which underrepresented groups might be struggling that are unique to them and their experiences …and what do we do then to reach these communities in a way that’s meaningful?” (P75)

#### 3.9.4. Subtheme iv: Recognizing and Understanding Weight Bias, Stigma, & Discrimination (29%)

Identified foci in this subtheme included needs for: (a) research investigating the forms, prevalence rates, and impacts of weight bias, stigma, and discrimination (4/14, 29%) and (b) recognizing implicit weight bias/stigma/discrimination in the eating disorder field (1/14, 7%).

“Weight bias affects researchers and clinicians in the field of eating disorders the same way it affects everyone, everywhere. …I thought I was not weight-biased, but somebody [who focuses on weight bias professionally] said, to me, …’well, of course, you have weight bias; everyone has weight bias,’ and … I’ve thought about it and [I] realize… I do [have weight bias] …it’s so much [a] part of the scenery …that you don’t even … recognize how much it affects how you perceive things. … I think there’s a lot of people who don’t recognize that.” (P38)

#### 3.9.5. Subtheme v: Taking, Understanding, and Accounting for the Narrative (21%)

Three participants called for taking, understanding, and accounting for the narrative of individuals with binge eating disorder (3/14, 21%), including: (a) identifying how to “listen for what people are telling us about their experience” (P60, 1/14, 7%) and (b) listening to- and understanding the unique experiences of individuals with binge eating disorder (1/14, 7%).

“… [I] think …we should be taking account of people’s narrative and life experiences and that should be informing our therapy and our therapeutic approaches.” (P93)

#### 3.9.6. Subtheme vi: Understanding Consequences of Binge Eating Disorder (14%)

Two participants identified a need to understand consequences of binge eating disorder and how they impact individuals with the disorder (2/14, 14%), including impacts on: (a) interpersonal relationships (2/14, 14%); (b) threat sensitivity (1/14, 7%); and (c) expression of sexuality and intimacy with partners (1/14, 7%). See statement by participant 60 in Table 9 and in Table 12.

## 4. Discussion

### 4.1. Analysis Results

The environmental factors that impact adult binge eating disorder continue to change and evolve, as does our awareness of them as a field. The themes identified here represent new and emerging areas of research and recognition pertaining to environmental factors that impact adult binge eating disorder pathology.

#### 4.1.1. Theme 1: Systematic Issues and Systems of Oppression (100% Expert Identification)

Expert recognition and literature findings suggest oppressive systems often intersect in ways that disproportionately impact specific populations. For example, findings suggest racial, ethnic, sexual, and trans/nonbinary gender minorities are: (i) more likely to experience homelessness [45], unemployment [46], poverty [46], and food insecurity [47,48,49]; (ii) more likely to experience and perceive discrimination [50,51,52,53,54,55,56,57,58] and stigmatization [56,59,60,61,62,63]; (iii) being directly targeted by tobacco-owned food and beverage marketing programs [64]; (iv) less likely to be included in eating disorder research [65] or screened by healthcare providers for an eating disorder [66,67]; (v) generally less likely to recognize the need for binge eating disorder treatment when present [68]; and (vi) less likely to receive treatment when needed [65,66,68,69,70,71,72]. Each of these factors have been independently linked to increasing risk for binge eating disorder [34,59,73,74,75]. It is likely that experiencing more than one factor exponentially increases the risk for developing binge eating disorder, specifically among racial, ethnic, and sexual minorities. However, more research is warranted to identify the extent to which these separate systems of oppression interact among specific populations of individuals, and the nature of their relationship with binge eating disorder.

Overall, expert recognition and emerging literature highlight the need for public education and policy change that can help reduce the oppression and discrimination inherent in these systems. Examples include: education and enforcement of equal eating disorder screening from healthcare providers across race, ethnicity, gender, sex, sexual orientation, weight- and socioeconomic status (as well as healthcare provider education on implicit biases and stigmatizations about binge eating disorder) [76]; inclusion of minority and marginalized populations in eating disorder research [65]; equal funding for binge eating disorder research relative to other disorders of similar prevalence [77,78]; as well as consensus on which agencies should provide such funding; and public education and policy change in food industry practices that target minorities and abet binge eating [64,79,80,81,82,83,84,85,86,87,88].

#### 4.1.2. Theme 2: Marginalized and Under-Represented Populations (100%)

Many themes herein highlight a paradigm shift from an old view that ascribes eating disorders to thin, affluent, white cis-gendered females (the “SWAG: skinny, white, affluent, girl” stereotype [89]) to a new recognition of populations of individuals with binge eating disorder who have been historically overlooked in the field. These populations include racial, ethnic, and sexual minorities but also extend to other non-white, affluent, cis-gendered female populations (e.g., males [68], individuals in “normally” sized/weighted/shaped bodies [68], and individuals with low socioeconomic status [34,47,68,73,90], especially those with past or present histories of food/nutrition insecurity [34,47,73,74,75,90,91,92,93], or use of government assistance programs like SNAP, food stamps, or welfare [47,75,91,93]). The experts recognize the historical mistake of overlooking these populations in research and clinically (theme 2) and recognize a need to include these populations going forward (themes 2, 9.iii,v).

Emerging literature supports the growing understanding that although white affluent cis-gendered females have traditionally saturated the treatment-seeking population (and thus research populations as a result), they do not exclusively represent all individuals who experience binge eating disorder. For example, studies have shown that 93.4–96.8% of individuals who meet DSM criteria for binge eating disorder never receive a formal diagnosis [68,94]; 67.3% do not perceive the need for treatment [68]; and 56.4–86.8% never receive or pursue treatment [2,68]. Moreover, individuals with eating disorder symptoms who are underweight, female, affluent, and/or white are more likely than their respective counterparts to perceive a need for treatment or receive a diagnosis or treatment [68].

The shortcomings in eating disorder recognition, screening, and treatment seem to affect specific minority populations more profoundly. For example, black, indigenous, and people of color have higher prevalence rates of binge eating disorder than their white peers [71,72,95], but make up <10% of participants in binge eating disorder research studies [65], are less likely to be screened by medical professionals for eating disorders [66,67], and are 50% less likely to be diagnosed or receive care [65,66,69,70,71,72]. Emerging literature suggests sexual minorities are at an increased risk for eating disorders [96], with a nationally representative U.S.-based study finding greater prevalence of adult binge eating disorder in non-heterosexual participants (2.2%) than in heterosexual participants (0.8%) [97]. Literature on prevalence rates among agender, transgender, and non-binary individuals focuses primarily on youth [47,59] but demonstrates transgender young adults also have higher prevalence rates of binge eating disorder than their cis-gendered counterparts (28% in females, 64% in males, and 73% in non-binary respondents) [59], with transgender males having higher eating disorder pathology scores than transgender females [47].

These minority and marginalized populations often experience multiple factors associated by the experts as relevant to binge eating disorder pathology (e.g., economic precarity, food/nutrition scarcity, stigmatization, and discrimination) [59,71,72]. For example, rates of food insecurity are significantly higher in Black and Hispanic households (vs. White households) [48,49]. Transgender and nonbinary individuals experience higher rates of family rejection [45] and homelessness [45], have three times higher rates of adult unemployment (15% vs. the 5% national average) [46], and over two times higher rates of poverty and food insecurity than the national averages (poverty: 29% vs. 12% national average [46]; food insecurity: 21.2% vs. 10.5% national average [47,48]). These populations also experience high prevalence of stigmatization [56,59,60,61,62,63], with higher levels of enacted stigma associated with higher odds of binge eating in transgender young adults [59]. More research is warranted to investigate other ways these independent factors may intersect to mediate or moderate the higher prevalence rates of binge eating disorder observed in minority populations.

#### 4.1.3. Theme 3: Economic, Food, and Nutrition Scarcity and Insecurity (93%)

Associations between household food insecurity or fluctuations in food availability and eating disorders [73,75,91,93,98] (including binge eating disorder [75,91,93]) are supported in the literature [30,31,32,34,35,47,68,71,73,74,75,90,91,92,93,98]. A 2020 systematic review and meta-analysis calculated binge eating disorder is 1.65 times more common in participants with food insecurity than in food-secure individuals (8.6% vs. 5.2% prevalence; *p* = 0.02) [75]. Studies also demonstrate receiving government assistance before 18 years of age (e.g., welfare, food stamps, SNAP) [74] or having low- and very low food security [93] are both associated with increased odds of having binge eating disorder (government assistance aOR = 2.95 [74]).

Studies conducted at a food pantry in San Antonio, TX between 2015–2016 [34,73,90] found 51.5% of respondents reported deliberately trying to limit food consumption or going >8 h without food consumption [73], which was significantly correlated with overall level of eating disorder pathology (*r = 0.25, p* = 0.0001) and higher BMI [73]. Three main reasons for food minimization were lack of resources, SNAP or food stamps being insufficient, and family reasons [73]. Examples included “minimizing effects of hunger for other family members (e.g., children), ‘stretching’ food to make it last longer, and prioritizing medical expenses,” [73].

These and other findings suggest limitations in current government assistance programs–or the way these programs are being used–may increase odds of recurrent binge eating, leading to binge eating disorder. While research continues to emerge on this relationship, some experts recognize this issue as systemic and likely requiring public education and policy change (themes 1–3). Additionally, it is important to include individuals with economic precarity and food scarcity/insecurity in research going forward, and to identify prevention and treatment options accessible to them.

#### 4.1.4. Theme 4: Stigmatization and Its Psychological Impacts (93%)

The experts and literature recognize internal and external stigmatization [56,60,61,62,63,76,99,100,101,102,103,104,105,106,107,108,109] and discrimination [50,51,52,53,54,55,56,57,58,110,111] as relevant to weight disorders [99,100,101,102,106], eating disorders [62,76,101,103,104,107,108], and binge eating disorder specifically [107,109]. Experts and literature alike recognize different forms of stigmatization (e.g., weight stigma [99,100,101,102,106], eating disorder stigma [109], mental health diagnosis stigma, and stigmatization around perceived eating behavior [109]). The literature also recognizes stigmatization among healthcare providers [76,105] and differences in stigmatization that may exist across gender [60,61], race/ethnicity [56,62], and sexual orientation [63]. Stigmatization is recognized in the literature as contributing to marginalization (a potential form of oppression) [62]. While some experts recognize stigmatization as being potentially traumatic, this possibility is not yet reflected in the literature, and holds potential for an important future research area.

#### 4.1.5. Theme 5: Trauma and Adversity (79%)

Expert awareness of the associations between trauma history or adversity and binge eating disorder are reflected in systematic reviews and meta-analyses collectively [19,112,113,114,115,116,117,118,119,120,121,122,123,124,125,126,127]. These forms of trauma can include adverse childhood experiences (ACEs), adverse life experiences (ALEs), family-related non-abuse ALEs, and post-traumatic stress disorder (PTSD). Similar associations have been observed in obesity [116,128,129], which has high comorbidity with binge eating disorder [2,130,131]. For example, Vincent Felitti’s hallmark ACEs Study [132] revealed major weight loss is perceived as “unsafe” and “threatening,” [133] and associated with high weight loss attrition rates (>50%) among individuals with obesity and comorbid ACEs [128,132,133,134].

Adverse childhood *food* experiences represent a specific form of adversity recognized by the experts that are also gaining recognition in the literature [74]. Coffino et. al. (2020) recognize childhood food neglect as “an ACE specially involving restricted food access” that increases odds of developing binge eating disorder (aOR: 2.95; 95% CI: 1.73–5.03]),” [74]. While the experts call for a greater understanding of the specific forms of trauma and adversity that individuals with binge eating disorder experience and the ways these factors interact with binge eating disorder pathology (themes 5, 9), it will also be important to evaluate existing literature that questions the internal validity of self-report measures of adverse childhood experiences [135,136].

#### 4.1.6. Theme 6: Interpersonal Factors (Threat & Threat Sensitivity) (64%)

“Interpersonal,” an adjective, refers to relationships or communication between people [137]. Experts and literature recognition of interpersonal factors as relevant to binge eating disorder pathology [138,139,140,141] are underscored by the success of interpersonal psychotherapy (IPT) in treating binge eating disorder [138,139,140,141,142,143,144,145,146,147]. National survey data collected between 2001–2003 [2] found 15.9% of individuals with binge eating disorder report severe impairment in their social lives, which exceeds reported prevalence rates for impairments in home-, work-, or personal lives [2]. Furthermore, individuals with binge eating disorder have significantly higher levels of interpersonal distrust [139], social insecurity [139]. attachment anxiety [139], and attachment avoidance [139] than non-clinical samples, as well as relational styles that are more domineering [141], cold [141], less- or non-assertive [138,140,141], more exploitable [140], and more socially withdrawn [138,141].

The lack of consensus among experts on the nature of the relationship between interpersonal deficits and binge eating pathology (e.g., directionality, possible mediators, moderators, and underlying mechanisms; Table 10) reflects an incomplete literature base. Ivanova et al.’s 2017 interpersonal model [138] found negative affect and affect instability mediate the relationship between interpersonal functioning and eating disorder psychopathology (*p* < 0.01–0.001) [138]. This model generally has the most support in the field [140,148,149] though incongruent findings exist [142,150].

Social ranking, social threat perception, and threat sensitivity have also been explored in the literature as being relevant to eating disorder pathology [151,152,153,154,155,156,157]. Work from Janet Treasure’s lab finds individuals with anorexia- and bulimia nervosa have heightened sensitivity to social rank-related cues (but impaired self-evaluation at an automatic level of processing) [153], genetically-rooted attentional biases to social threat [152,155] (but deficits in emotion recognition and regulation) [152], and cognitive avoidance of threat-related information [151,155], Currently, these findings remain largely untested [158] or unreplicated [157] in individuals with binge eating disorder.

Given the experts’ comments on the traumatic ways in which individuals with binge eating disorder experience a variety of forms of stigmatization and discrimination—and the ways these experiences often extend to an individual’s value and worth (theme 4)—it may not be surprising that individuals with binge eating disorder struggle with interpersonal communication and relationships. Possible connections between stigmatization, social bullying, adverse interpersonal experiences, and interpersonal deficits or even post-traumatic stress around interpersonal adversity certainly warrants future investigation.

The theme of social support is one that bridges the themes of interpersonal factors (**theme 6**) and social media engagement (theme 7). Literature shows individuals with binge eating have significantly greater levels of loneliness and lower levels of perceived social support relative to individuals without loss of control eating (*p* < 0.001) [159]. Furthermore, greater levels of social support are significantly associated with reductions in binge eating [139,160,161]. However, emerging literature demonstrates that not all forms of social support are equal for all individuals with binge eating disorder. For example, social support from family is associated with less likelihood of binge eating in African American women but greater likelihood of binge eating among white women [161]. A Chinese analysis of eating disorder treatment experiences further concluded “parents, friends, and partners were sources of social support, but participants largely felt misunderstood or blamed by these same entities,” [162]. Overall, the literature supports participant 84′s statement that “other people can be part problem as well as part of the solution.” The impacts of different forms of social support on different populations of individuals with binge eating disorder warrants further investigation, as does the nature of the relationship between interpersonal factors and binge eating (e.g., regarding directionality, existence of mediators, moderators, and underlying mechanisms).

#### 4.1.7. Theme 7: Social Messaging and Social Media (50%)

The experts’ recognition of negative impacts of social media use on binge eating disorder is reflected in the literature. In Chinese children, each additional hour of total screen time, social networking, texting, and/or watching/streaming television shows/movies per day is prospectively associated with significantly higher odds of binge eating disorder [163]. Increased photograph investment and investment in others’ “selfies” on social media have both been associated with significantly increased odds of meeting criteria for binge eating disorder among Australian adolescents [164]. A 2022 integrative review of social media use and binge eating further concluded “the more participants use social media, the more likely they are to have increased appetite or intention to eat, which can lead to binge eating,” [165].

Media messaging was perhaps one of the first factors explored as contributing to eating disorders broadly [166,167,168], supported by neuroimaging findings of neurobiological alterations related to body comparisons and distortions in individuals with anorexia nervosa [169,170,171]. Studies in binge eating disorder support the presence of body image disturbances (e.g., body weight/shape overvaluation), dissatisfaction, and body-related cognitive biases (but with accurate ratings of body shape) in individual with binge eating disorder [172]. However, the direct culpability of media-endorsed body images remains to be tested. Giordano (2015) suggests: “considering the portrait of thin models in the media industry as responsible for eating disorders is a misanalysis of the problem,” and states, “media images, the public preference for extreme thinness, are, like eating disorders themselves, the expression of normative values” that relate thinness with “self-control, discipline, and austerity”, [167].

While it may be true that sociocultural associations exist between thinness and self-control (and—by default—between larger body sizes or loss of control eating and lack of self-control or discipline), these associations warrant investigative testing. The themes identified here suggest binge eating disorder is often associated with hardships that are outside of the control of the individuals who experience them (e.g., systematic oppression, marginalization, stigmatization, trauma, and adversity). The ways in which an individual’s lack of control of *external* processes relates to lack of control of *internal* processes (and loss of control eating) warrant further investigation, as does an assessment of self-efficacy among individuals with binge eating disorder.

Overall, experts and literature suggest binge eating is often simultaneously normalized and stigmatized socioculturally. Sociocultural messages around binge eating and larger body sizes extend from advertisements that promote and glamorize food consumption while also reinforcing a drive for weight loss and thinness to Instagram selfies and tweets that emphasize thinness and conceal lack of control or hardships. An assessment of these messages may help explain why binge eating disorder itself is so minimalized [105], marginalized, and overlooked, despite its prevalence and gravity.

The experts’ recognition of social media as an inadequate replacement for community and/or solution to loneliness and isolation (theme 6, e.g., P72) also bears further investigation. Despite many negative associations between social media engagement and binge eating disorder development [163,164,165], Nutley et al.’s qualitative analysis of Reddit use (www.reddit.com) among individuals with eating disorders during the COVID-19 pandemic concluded that many individuals used Reddit forums for support during the pandemic [173]. To this end, a variety of literature shows statistically significant associations between social support and reductions in binge eating [139,160,161]. Further investigation is warranted to determine the ways in which social support differ when achieved through different forms of online social mediums (e.g., engagement in Instagram/Facebook/Twitter vs. support-oriented Reddit forums or Facebook groups), online or in-person treatment-oriented mediums (e.g., treatment teams, professional-led support groups, or peer-led support groups such as overeaters anonymous), or in-person (e.g., in-person support from family and friends).

#### 4.1.8. Theme 8: Predatory Food Industry Practices (29%)

Minority expert recognition of food industry practices designed to manipulate consumer choices (29%) is proportionately under-represented in peer reviewed journals and in the popular press [64,79,80,81,82,83,84,85,86,87,88]. Experts and literature recognize that food environments and manufacturers can (and do) leverage food decisions by influencing variables known to control consumption that can also result in binge eating (e.g., marketing, packaging, container shape, portion size, product salience, and hedonic factors like salt, fat, sugar, and structure/texture) [80,86]. Literature also recognizes similarities in deceptive tobacco- and food industry marketing practices (e.g, intentionally emphasizing personal responsibility, paying research scientists to criticize opposing research as “junk science,” lobbying with “massive resources to stifle government action,” “manipulating and denying …the addictive nature of their products,” and marketing to children to capture lifelong consumers) [85]. More recent findings identify direct tobacco industry involvement in food marketing strategies [64,87] that were not recognized by the experts. Examples include R J Reynolds’ ownership of Hawaiian Punch [87], Philip Morris’ ownership of Kool-Aid [87], Capri Sun [87], Tang [87], and General- and Kraft Foods [64], and Phillip Morris’ direct transfer of “expertise, personnel, and resources from its tobacco to its food subsidiaries, creating a racial/ethnic minority-targeted food and beverage marketing program modeled on its successful cigarette program,” [64]. This depth of tobacco- and food industry involvement highlights some of the ways oppressive tobacco and food systems directly target racial and ethnic minorities, possibly contributing to the systemic oppression that can increase the risk for developing binge eating disorder.

Literature–but not experts–also identifies food industry efforts to influence nutrition research and professional activities [82,83] through forming and funding “science-sounding” research institutes and journals used to develop and publish industry-supported studies and industry-benefiting guidelines on scientific integrity, conflict of interest, and public-private partnership, thus shaping the nature of the science that drives public policy to its benefit [82]. These practices are important to address because they influence food consumption in ways that can result in binge eating and specifically because they are *not* recognized by the majority of experts, suggesting they may not be known. Thus, the experts’ call for public education and policy change around food industry practices (theme 6) seems particularly imperative.

#### 4.1.9. Theme 9: Accounting for Narratives & Life Experiences through Open-Ended Research

The themes identified herein emphasize the need for open-ended research that can access and include the ~70% of individuals who experience binge eating disorder symptoms but may not recognize their need for treatment [68] as well as the ~95% of individuals with binge eating disorder who never receive a formal diagnosis [68,94] and the ~56-87% who never pursue or receive treatment [2,68]. It is equally important to identify research recruitment and dissemination tactics and treatment options that include minority and marginalized populations and receive and account for the unique treatment barriers these populations face. Specifically, these themes emphasize the value of taking and accounting for the narratives and life experiences of *all* individuals with binge eating disorder to better inform our current understanding of recurrent binge eating pathology and the systemic factors that impact it, as well as the unique challenges that historically underrepresented groups face. The experts recognize this need (**theme 9**), which is reflected in the literature [47,65,70,71,73,75] and shared by the authors.

### 4.2. Expert Demographics

While this study’s sample provides an accurate demographic representation of eating disorder experts (92% white, 100% not Hispanic or Latino), it does not accurately represent the demographic profile of individuals who experience adult binge eating disorder, which has higher rates of Hispanics, Latinos and Blacks, Indigenous, and Peoples of Color [71,72,174]. The demographic discrepancies between those who study and treat adult binge eating disorder and those who experience it are not insignificant. These discrepancies highlight the importance of including marginalized populations in academic and clinical training opportunities for adult binge eating disorder research and care, and of emphasizing community- and narrative-based approaches to research.

Another important limitation (and oversight) of this study is that demographic data was collected for sex assigned at birth, but not for gender. This is a shortcoming for two reasons: (1) gender is relevant to this study question and sex assigned at birth is not, and (2) asking for sex assigned at birth follows an old convention (collecting information on sex assigned at birth by default rather than collecting only the most relevant information for the study question) that fails to include and account for equity and diversity. While the question of sex assigned at birth is not relevant to this study question, the question of gender is. It would also have been relevant and important to identify whether any study participants identify as agender or transgender since binge eating disorder has higher prevalence among transgender and gender non-binary individuals (at least in youth and young adults; to the authors’ knowledge we are still lacking this information in adults) [47,59]. Overall, the field should work toward developing a base of researchers and clinicians who study and treat binge eating disorder that more accurately represents those who experience it.

### 4.3. Study Limitations

This research includes a small sample of individuals (*n* = 14). Thus, we cannot identify how accurately the themes and views identified here represent those of all binge eating disorder researchers, clinicians, and administrators. Moreover, NIH R01, T32, or P grant funding is one of four possible eligibility criteria that researchers were required to meet for participation eligibility (Table 3). Eligible participants were required to meet *one* of the four criteria presented in section I of Table 3 (not all four). Thus, our study did not limit researcher participation to U.S. researchers. However, this criterion did present a bias for inclusion of academics and researchers within the U.S. Although the study sample does include individuals from the UK, AU, and CA, 50% of study participants are American. Thus, the themes identified here must be interpreted with caution when done in the context of binge eating disorder pathology globally (vs. within the U.S.). Furthermore, the qualitative analysis of expert interviews were conducted by two individuals (BB and HZ with aid of CB). Thus, we cannot assess how accurately the themes identified here represent the true themes valued by expert binge eating disorder researchers (including those in this study and those at large). These limitations are standard in the field of reflexive thematic qualitative analysis and are not generally viewed as discounting the methodology as a whole [29].

It should also be noted the findings reported here come from a larger study that collected information from the fourteen experts reported here on their perspectives of adult binge eating disorder pathology and treatment broadly. Because of the robust commentary regarding binge eating disorder as a social justice issue, the authors felt the theme was substantive enough for a separate manuscript, even if not the exclusive focus of the interviews. If the interviews had focused solely on environmental factors relevant to adult binge eating disorder pathology, perhaps we could have delved further into the social justice issues deemed relevant to adult binge eating disorder.

### 4.4. Study Strengths

The study of social justice issues relative to their influences on behaviors is a relatively new contribution to “hard” science. This movement provides a new lens through which to view eating disorders. Our study aims to broaden the awareness of social justice issues that are both important and historically overlooked in the field of adult binge eating disorder. To the authors’ knowledge, this study is also the first to synthesize expert opinion on environmental factors contributing to adult binge eating disorder pathology. The study’s use of systematic inclusion criteria (Table 1) helps ensure as accurate population representation as possible of expert binge eating disorder researchers, clinicians, and healthcare administrators. A diverse group of experts were recruited internationally for the study and the study sample includes a balance of binge eating disorder experts involved in the field at all levels, including researchers (PhDs/ScDs, MD/PhDs), medical doctors (MDs, B\MBChBs), licensed therapists (LPs, R. Psychs, FRCPsychs), licensed and registered dieticians (LDs, RDs and certified eating disorder registered dieticians, CEDRDs), intuitive eating specialists, healthcare administrators, and public health and policy advocates (MPH) (Table 3). The study’s clinical criteria include clinicians with high clinical and academic affiliations as well as those likely to be most accessible to individuals with binge eating disorder (e.g., most commonly identified through a google search or through reading a popular press book on binge eating disorder). The latter criteria are met by including clinicians with high popular press distinction and online directory listings. Thus, this study includes clinicians who are active in the clinical space both academically and commercially. The study’s academic inclusion criteria ensure academics with the greatest amount of funding and publication output both recently and historically are included. Thus, the academics included in this study are those who drive and influence the field and its literature base. Although the NIH funding criteria presents a bias for inclusion of academics within the U.S., the study sample does include a balanced geographic representation, with individuals across the U.S. (East Coast, Midwest, and West Coast), as well as in countries outside of the U.S.

## 5. Conclusions

Interviewed experts conveyed a growing awareness of systemic issues related to binge eating disorder. These issues include systems that disproportionately affect marginalized and minority populations who have historically been overlooked, under-diagnosed, and under-treated in the field. Examples of these issues include predatory food industry practices and food environments, body weight/shape/size discrimination, insurance and healthcare systems, and the eating disorder research field and eating disorder funding agencies. Interviewed experts recognize a need for public policy changes that take responsibility for changing systemic factors influencing binge eating disorder (e.g., food industry practices and food environments). The experts also recognize the need for including these populations in research, and for “taking account of people’s narrative[s] and life experiences” to better inform our current understanding of binge eating disorder and the systemic factors that impact it.

## Figures and Tables

**Figure 1 ijerph-19-06243-f001:**
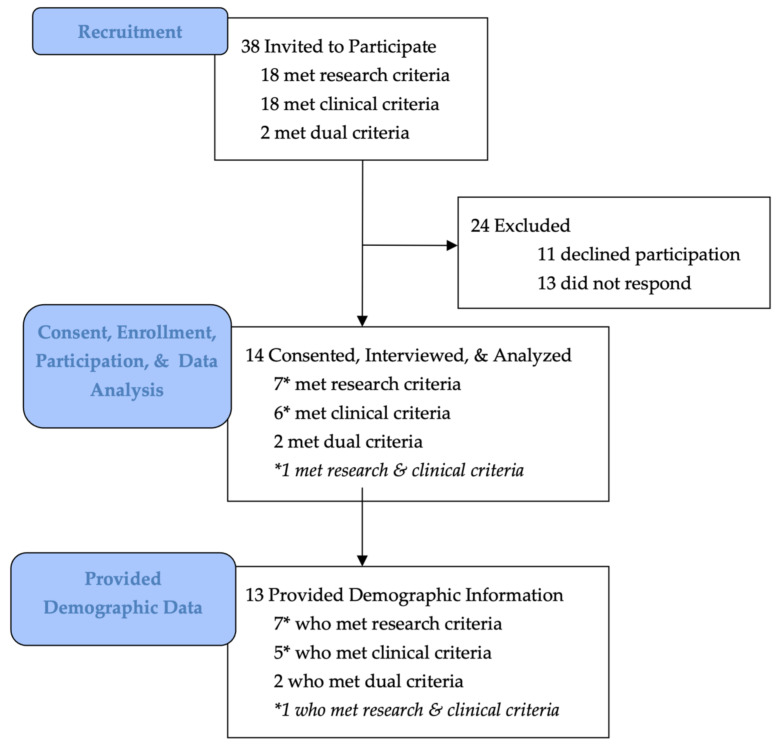
Diagram of study flow, from participant identification to enrollment and follow-up. Thirty-eight experts met enrollment criteria and were invited to participate in the study. This included 18 experts who met the academic/research criteria (18/38, 47%), 18 experts who met the clinical criteria (18/38, 47%), and two who met the dual criteria (2/38, 5%; Table 1). Fourteen eligible experts consented, enrolled, and participated in the study (14/38, 37%), including six individuals who met the academic/research criteria (6/14, 43%), five who met the clinical criteria (5/14, 36%), one who met both the academic/research and clinical criteria (1/14, 7%), and two who met the dual criteria option (2/14, 14%). Thirteen participants (13/14, 93%) provided demographic information and were included in demographic analysis (Table 3). All 14 participant interviews were included in thematic analysis.

**Table 1 ijerph-19-06243-t001:** Participant eligibility criteria and number of participants who were recruited, and who consented and enrolled.

**I. Eligibility criteria for researchers (18 recruited, 7 enrolled)**
Eligibility criteria for researchers required meeting one of the following four criteria (I.1–4):
1. ≥1 active R01, T32, or P grant on binge eating or food addiction Identified on NIH RePORTER (https://report.nih.gov; accessed on 5 Januuary 2021) using the search terms: ”binge eating,” “binge eating disorder,” “food addict,” “food addiction,” or “food addict OR food addiction.” Identified grants containing the following F9 searches terms were included: “binge eating disorder,” “BED,” “binge eating,” “BE,” “compulsive eating,” “CE,” “overeating,” “OE,” and “food reward,” as well as “loss of control,” “obesity,” and “eating disorder” when relevant to binge eating disorder. Exclusion criteria: (A) grants not relevant to binge eating disorder, binge eating, compulsive eating, or overeating; (B) grants pertaining to anorexia nervosa or bulimia nervosa, without also pertaining to binge eating disorder; (C) grants pertaining to childhood or adolescence without also pertaining to adult binge eating disorder.
2. Last author of ≥10 PubMed publications published 2010–2020 AND ≥5 PubMed publications in 2015–2020 relevant to binge eating disorder Retrieved January 2021 with the search term “binge eating disorder” in adults, young adults, and middle aged.
3. Last author of ≥5 PubMed publications published in 2015–2020 relevant to food addiction ^a^ Retrieved January 2021 with the search term “food addiction” in adults, young adults, and middle aged.
4. Referral from someone who meets one of the qualifications above (I.1–3).
**II. Eligibility criteria for clinicians and healthcare administrators (18 recruited, 6 enrolled)**
Eligibility for clinicians and healthcare administrators required meeting ≥3 of the following 8 criteria:
1. Association of Eating Disorders (AED) Award Winner or Honoree in 2010–2020 [22]
2. Castle Connolly Top Doctors Distinction in Psychiatry—Eating Disorders (2020/21) [23]
3. Executive position/board member for one of ten relevant societies: Academy of Nutrition & Dietetics Academy of Eating Disorders (AED, FAED) [24] American Society for Metabolic and Bariatric Surgery (ASMBS) Behavioral Health Nutrition Society Eating Disorder Research Society (EDRS) International Association of Eating Disorder Providers (IAEDP) Johns Hopkins 2020 Eating Disorders Conference National Center of Excellence for Eating Disorders (NCEED) National Eating Disorder Association (NEDA) Obesity Society
4. Adult binge eating disorder provider listed in the Provider Directory for the National Eating Disorder Association (NEDA) [25] or the Alliance for Eating Disorders Awareness [26]
5. Association with an eating disorder program or treatment center with five or more locations listed in the NEDA Directory
6. Popular press distinction, including: Recognition in a 2016 New York Times article on eating disorder treatment centers [27] Author of one of 15 eating disorder books recognized in a 2021 posting on a self-help website for binge eating [28]
7. Referral from an individual who meets ≥2 other qualifications (II.1–6)
8. Registered Dietician (RD) or Certified Eating Disorder Registered Dietician (CEDRD) meeting 2 other criteria above (II.1–7)
**III. Additional Eligibility Criteria (2 recruited, 2 enrolled** ^b^
In addition to the academic/research and clinical criteria above, individuals who met ≥1 academic/research criterion (I.1–4) and ≥1 clinical criteria (II.1–8) were also eligible.

Results expressed as *n* (%). Percentages expressed as *n*/14 times 100. BED: binge eating disorder. ^a^ Unlike the criteria for binge eating disorder publications (I.2), this criterion did not require more than ten publications in the past ten years because the concept of food addiction is still relatively new and has developed mostly in the last eight to ten years. ^b^ Both participants each met two academic/research criteria and two clinician/healthcare administrator criteria. Refs. [22,23,24,25,26,27,28] refer to citations in References.

**Table 2 ijerph-19-06243-t002:** Interview questions pertaining to environmental factors relevant to adult binge eating disorder pathology.

** *Question* **	** *n asked (n/14)* **
Please describe your perspective on (or knowledge of) literature and research findings, current clinical guidelines, and your own personal experiences that relate to binge eating disorder pathology and treatment.	14 (100%)
How do you view the disease process in relation to the following possible aspects, and how important is it for treatment interventions to address these aspects (if at all)?	
Physical/Biological Cognitive/mental Emotional Spiritual Economic Social Cultural Other	14 (100%) 14 (100%) 14 (100%) 14 (100%) 11 (79%) 12 (86%) 12 (86%) 14 (100%)
How do you view the disease process in relation to the following possible aspects, and how important is it for treatment interventions to address these aspects (if at all)?	
Physical/Biological Cognitive/mental Emotional Spiritual Economic Social Cultural Other	14 (100%) 14 (100%) 14 (100%) 14 (100%) 11 (79%) 12 (86%) 12 (86%) 14 (100%)
Please describe your view on the following health factors as they relate to adult binge eating disorder pathology and treatment:	
Weight Stigma Malnutrition/Undernutrition Sleep Early Life Trauma/post-traumatic stress disorder (PTSD) Coronavirus-19 (COVID-19)/Quarantine Other	9 (64%) 10 (71%) 11 (79%) 9 (64%) 10 (71%) 12 (86%)
Are there any other aspects of binge eating disorder pathology that you feel are important to address or discuss (that have not been addressed above)?	12 (86%)
Please describe your perspective on current research gaps that exist in the field of binge eating disorder.	14 (100%)
Do you have any other suggestions that relate to future research on binge eating disorder?	14 (100%)

Results expressed as *n* (%). *n* = number participants asked. Percentages expressed as *n*/14 times 100.

**Table 3 ijerph-19-06243-t003:** Characteristics of study participants (*n* = 14) *.

**Accreditations**
Fellow of the Academy of Eating Disorders (FAED)	8 (62%)
Doctor of Philosophy (PhD) or Science (ScD)	8 (62%)
Medical Doctor (MD)	4 (31%)
Licensed or Registered Dietician (LD/RD) or Registered Dieticians Certified in Eating Disorders (CEDRD)	4 (31%)
Healthcare Administrator	2 (15%)
Certified Chef	1 (8%)
Certified Intuitive Eating Specialist (CIES)	1 (8%)
Fellow of the American College of Neuropsychopharmacology (FACNP)	1 (8%)
Bachelor of Medicine Chirurgical Doctor (Bachelor of Surgery) (B\MBChB)	1 (8%)
Master’s in Public Health (MPH)	1 (8%)

**Sex (at birth)**
Female	8 (62%)
Male	5 (38%)
Other	0 (0%)

**Age**	
55 ± 10.2 years (range: 37–44 yrs., *n* = 13)	

**Ethnicity**	
Hispanic or Latino	0 (0%)
Not Hispanic or Latino	13 (100%)

**Race**	
American Indian or Alaska Native	0 (0%)
Asian	1 (8%)
Black or African American	0 (0%)
Native Hawaiian or Other Pacific Islander	0 (0%)
White	12 (92%)
More than one race	0 (0%)

**Geographical location of residence**	**7 reported**
United States of America (USA)	5 (71%) **
United Kingdom (UK)	1 (14%) **
Australia (AU)	1 (14%) **
Canada (CA)	1 (14%) **

**Eligibility criteria met:**	
Research/Academic	6 (43%)
Clinical/Administrative	5 (36%)
Both (Research/Academic AND Clinical/Administrative)	1 (7%)
Combined (≥1 Research/Academic and ≥1 Clinical Administrative)	2 (14%)

Results expressed as *n* (%) or mean ± SD. * Percentages are expressed as *n*/13 times 100, as one participant did not provide demographic data. ** Percentages are expressed as *n*/7 times 100, as only seven participants provided this data.

**Table 4 ijerph-19-06243-t004:** Participant responses consistent with Theme 1, “systematic issues and systems of oppression”.

**Identified Systems of Oppression that Relate to BED Pathology**	**14 (100%)**
Systematic discrimination	12 (82%)
Body weight/shape/size discrimination (see Theme 4)	12 (82%)
Structural racism	2 (14%)
Structural sexism	1 (7%)
Media messaging and sociocultural ideals/mandates (see Themes 4 and 7)	12 (82%)
Perpetuating stigmatization	12 (82%)
Body weight/shape/size ideals (and discrimination)	12 (82%)
“Diet culture”	3 (21%)
Movement & fitness ideals	2 (14%)
Insurance and healthcare systems	9 (64%)
Insurance costs and coverage	6 (43%)
Treatment costs	6 (43%)
Systematic stigmatization from healthcare providers	6 (43%)
Geographical access to treatment resources	4 (29%)
Mandated movement for individuals in larger bodies	2 (14%)
Provider scarcity	1 (7%)
“Predatory” food industries/environment (see Section Theme 8)	4 (29%)
Abuse (sexual, emotional, or physical)	4 (29%)
Geographical systems ^1^	4 (29%)
) Eating disorder research as a field ^2^	3 (21%)
) Eating disorder research funding ^3^	2 (14%)
Economic exploitation ^4^	1 (7%)
School systems	1 (7%)
Legal systems	1 (7%)
Police harassment	1 (7%)
**Additional participant statements regarding systems of oppression:**	
*“In [my country], there is no public funding for people who have binge eating disorder. …they’re just sort of on their own when it comes to treatment.” (P38)*	
*“…the food industry … there’s all these food scientists and psychologists that go work for this industry to figure out how to generate food that is the most profitable and the hardest to not overeat. …there are companies and individuals profiting off of making food that will lead someone down a path to binge eating disorder or binge eating… If that’s not a system of oppression I don’t know what is.” (P16)*	
*“There’s so much less research on binge eating disorder than [on] anorexia nervosa or bulimia nervosa … how it’s experienced across different communities, how intersectional systems of oppression affect risk of developing it…” (P16)* *“How we think about eating disorders is that … anorexia was kind of the granddaddy, … the thing we knew first, and then bulimia kind of grew out of that next, and … people used to refer to [it] as … failed anorexia… … And I would say, in part that, … binge eating disorder… was … thought of … Initially [as being] like bulimia, but without the purging …so [the] same mechanisms… …[and] I would say traditionally, when I’m talking to my colleagues in the eating disorder field… the dominant mechanisms that … have a tendency to be most thought of are things like restraint and shape and weight overvaluations but there’s starting to be a bigger push to … have a more encompassing view on mechanisms like reward and inhibitory control and emotion, distraction, things like that.” “… I think in part because the restraint stuff wasn’t necessarily panning out with binge eating disorder quite as well.” (P19)*	
*“I think binge eating disorder is in a little bit of a strange place in the United States. Because NIH you have mental health conditions in one place at NIMH, and you have metabolic conditions at another place at NIDDK. And sometimes I think binge eating disorder doesn’t have a home. So how many RCTs have been funded by the NIH in the treatment of binge eating disorder? … I think that there’s a professional socio-cultural administrative framework that is having trouble getting its arms around binge eating disorder…” (P33)*	
*“[I am] constantly surprised at how easy it is to get money for [weight disorders in comparison to eating disorders]. The weight loss trial, not hard at all [to get funded]. If you think you’ve got the diet that works, it’s not hard to persuade governments or philanthropists or people to fund you. It’s very hard to persuade governments or philanthropists to fund if we think we’ve got some new eating disorder treatment, or some enhancement of initial treatment, we find it much harder to get [funding for] than yet another diet.” (P93)* *“Systems of oppression, which actually work through [many] domains [e.g., emotional, spiritual, economic, social, and cultural]… if you think about structural racism or … structural sexism or economic exploitation or there’s all these kinds of systems, they might be legal systems or economic systems that will increase risk for different kinds of conditions and certainly binge eating disorder is affected by all of this—discrimination, housing precarity, economic precarity, all of these the way these systems will affect people and families and communities… … …how structural racism might set people up on a path to end up facing food insecurity or nutrition insecurity … schools failing young people, maybe somebody doesn’t get their degree or [has] other things going on in their communities where they’re being harassed by police … or they’re being abused and nobody’s watching [or] nobody’s there to protect them. There are so many levels of ways that systems are failing people—particularly in children [and] particularly in communities that are marginalized—that then create ACES [adverse childhood experiences] … that could increase the risk of binge eating disorder.” (P16)*	

Results expressed as *n* (%), in which percentages are *n*/14 times 100. ^1^ Defined as a relationship in economic wealth distribution wherein a worker does not receive proper compensation for his/her work (Arnold, 1995), e.g., from an employer or with a spouse. ^2^ E.g., geographical inequities in provider and treatment access, and in the Supplemental Nutrition Assistance Program (SNAP) that can limit its effectiveness (Levine 2018; Hoynes et al., 2022; Ziliak 2016) treatment access. ^3^ E.g., operating from an anorexic-centric perspective/understanding. ^4^ E.g., lack of funds for eating disorder research relative to disorders of similar prevalence lack of clarity regarding what agencies should fund eating disorder research. Abbreviations: BED, binge eating disorder; NIDDK, National Institute of Diabetes and Digestive and Kidney Diseases; NIH, National Institute of Health; NIMH, National Institute of Mental Health.

**Table 5 ijerph-19-06243-t005:** Participant responses consistent with Theme 2, “marginalized and under-represented populations”.

**Marginalized & under-Represented Populations**	**14 (100%)**
Low socio-economic status/economic insecurity (see Theme 3.i)	13 (93%)
Food or nutrition scarcity (see Theme 3.ii–iii)	10 (71%)
Male sex/gender	8 (57%)
Racial and ethnic minorities (e.g., BIPOC)	5 (36%)
Lesbian, gay, bisexual, and transgender, queer (LGBTQ) & nonbinary	3 (21%)
Age	2 (14%)
Religion	1 (7%)
**Additional participant statements on minority- and marginalized populations**	
*“The number of people that I’ve seen and done evaluations on [who] are really surprised to learn that the way that they’ve been eating is actually considered disordered, and that they have an eating disorder, and I think that that’s especially true for men, I think that’s especially true for any individuals [who] don’t fit that stereotypical mold of who has an eating disorder. …We know that unfortunately eating disorders have been hampered by these old stereotypes about who’s affected, and that leaves millions of people undetected with an eating disorder. …There’s a lot of emphasis these days on making sure that we’re meeting the needs of underrepresented groups and so a lot of people are talking about how eating disorders don’t discriminate, and that’s certainly true, and I think it also makes sense to talk about the specific ways in which underrepresented groups might be struggling that are unique to them and their experiences versus just saying ‘anyone can be affected.” (P75)*	
*“So much of the eating disorder perspectives and history … that we attend to are very female-focused, … and come out of … the female gender orientation. …I think anorexia [nervosa] kind of set the stage [for a current understanding of eating disorder pathology and treatment], [and anorexia nervosa] is so dominantly female.” (P16)*	
*“Certainly, there has been discussion in the eating disorder world … about whether different ethnicities have different levels of acceptance of overweight and obesity. So, one wonders whether that has impacts on … the frequency of the distress about binge eating disorder or the wish for treatment.” (P46)*	

Results expressed as *n* (%), in which percentages are *n*/14 times 100. Abbreviations: BIPOC, black, indigenous, and people of color; LGBTQ, lesbian, gay, bisexual, transgender, and queer.

**Table 6 ijerph-19-06243-t006:** Participant statements consistent with Theme 3, “economic, food, and nutrition scarcity and insecurity”.

**Subtheme (i) Economic Aspects of Binge Eating Disorder**	**13 (93%)**
Direct connections between BED pathology and economic status/precarity	5 (36%)

**Potential mediators and moderators of relationship between economic status/precarity and BED pathology**	**9 (64%)**
Food insecurity	5 (36%)
Nutritional access/insecurity	5 (36%)
Food environment	3 (21%)
Mental health risks	2 (14%)
COVID-19 pandemic	2 (14%)
Access to treatment resources	2 (14%)
Weight biases & descrimination ^1^	1 (7%)

**Subtheme (ii) Topics related to food insecurity & scarcity:**	**9 (64%)**
Potentially disrupting one’s relationship with food or eating	5 (36%)
Linked to economic insecurity	5 (36%)
Cited research findings linking food insecurity to BED ^3^	5 (36%)
Increasing risk for other physical and psychological health problems	4 (29%)
Linked to the COVID-19 pandemic	2 (14%)
Childhood adverse food experiences as important ACEs ^2^	1 (7%)

**Subtheme (iii) Topics related to nutrition scarcity:**	**6 (43%)**
Linked to lower socioeconomic status	4 (29%)
Linked to food environment	3 (21%)
Cited research findings linking nutrition scarcity to binge eating and obesity4	1 (7%)
Cited research relating urbanization factors to increased risk for BED5	1 (7%)

**Additional participant statements regarding economic insecurity:**	
*“The big thing about economics and binge eating disorder is that horrible availability of foods. [The foods that are accessible to lower income individuals] are really … great binge foods. They’re not great nutritional foods. … I can buy a half dozen quarter-pounders for what it might cost to get a decent meal. [so] …it’s not that big [of] a surprise that if I haven’t got the money, I go and buy a couple of quarter-pounders rather than trying to pay for a meal that I can’t pay for…” (P84)*
*“If you have less money, if you know lower socioeconomic status, then you are maybe forced to eat less good food, less healthy food, maybe food with… less good fat content [and] that then could…—if you have a biological vulnerability—change your brain more than somebody else who eats healthier, and that then might … flip the switch to then engage more in [binge eating] behaviors…” (P53)*

**Additional participant statements regarding food insecurity:**	
*“Recent data have come out between 2017 and 2020 around food insecurity, and the higher rates of [binge eating disorder] among people with food insecurity [but] we as a field, I don’t think, have paid enough attention to food insecurity [or] screening for food insecurity [or] addressing it in our client population, much less talking to the food insecurity world about addressing binge eating disorder, in particular among people with food insecurity.” (P60)* *“Access to food is a big, big deal. …In households where there’s …food scarcity, [that] can lead to binge eating. You don’t know when you’re getting your next meal? And it’s in front of you? And you’re really, really, really hungry because you haven’t eaten in a while. And then there’s food around? What do any of us do when we’re really hungry? We eat.” (P7)*
*“You can’t underestimate the impact of [food insecurity] on somebody’s eating disorder. If [an individual has] food insecurity [gets] a lump sum of financial resources or food resource over a period of time and that’s supposed to last them over a week or a month, and they have binge eating disorder, it’s not unlikely that a significant portion of that food may be consumed and then they don’t have resources to buy more. And so now we have somebody who’s managing a binge with the financial constraint, which may likely add to the guilt of that eating disorder behavior and feel like it’s all their fault, [which can lead] back to the sort of cognitive thoughts [of], ‘I’ll never be able to do this. This is all my fault. I have no willpower. I’m a terrible person. How come I can’t do this?’ So, I think economically, we really have to pay attention to the impact of economic status, but particularly around food insecurity. And when you look at the data on financial hardship, food insecurity is often the highest ranked area of struggle, right? There’s housing, there’s medical, there’s utility, and then there’s food and food is the one that people express most frequently struggling with when they have financial hardship.” (P60)*

**Additional participant statements regarding nutrition scarcity:**
*“We have … whole … city areas that are geared up around fast food. … [there are places where you can] very easily get a hold of fast food. You [can’t] very easily get a hold of decent food.” (P84)*
*“[Undernutrition] is an aspect of any eating disorder because even in binge eating disorder, you’re going to find people with malnutrition. Malnutrition doesn’t discriminate. [There is] research on malnutrition [showing that] a lot [of individuals with malnutrition] [are] in a larger body. So that aspect, we know that malnutrition can affect our food preoccupation.” (P37)*

Results expressed as *n* (%), in which percentages are *n*/14 times 100. ^1^ E.g., individuals in larger bodies experience economic discrimination. ^2^ adverse childhood experiences that are often overlooked and under-screened, but that potentially relate to adult eating disorder pathology. ^3^ Two specifically [34,35] and two broadly. ^4^ Three separate areas of research demonstrate that: (a) malnutrition can occur in individuals with larger bodies, (b) malnutrition can lead to food preoccupation [41]; and (c) maternal malnutrition is linked to offspring obesity (e.g., Although the participant mis-referenced Aamodt, 2016 [43]—which cites Tripicchio et al., 2014 [44]—Parlee et al. [42] note the Dutch famine study found gestational maternal malnutrition increases odds of offspring adult obesity (Ravelli et al., 1976, 1998, 1999 in Parlee et al., 2014) and animal studies find maternal nutrient-or protein deficiency causes adult offspring obesity (see citations 57–78 in Parlee et al., 2014) [42]. ^5^ E.g., food and nutrition insecurity and poverty [35]. Abbreviations: ACEs, adverse childhood experiences; BED, binge eating disorder; COVID-19, Coronavirus-19.

**Table 7 ijerph-19-06243-t007:** Participant statements consistent with Theme 4, “stigmatization and its psychological impacts”.

**Subtheme (i) Forms of Stigmatization Recognized as Relevant to BED**	**13 (93%)**
Body weight/shape/size stigmatization and discrimination	12 (82%)
Eating disorder diagnosis stigmatization	5 (36%)
Mental health diagnosis stigmatization	5 (36%)
Any medical diagnosis stigmatization	1 (7%)
Stigmatization around perfectionistic food/eating ideals	1 (7%)
*These stigmatizations suggested as having higher prevalence in specific populations* ^1^	2 (14%)

**Subtheme (ii) Body weight/shape/size stigmatization described as:**	**5 (36%)**
Potentially exacerbating BED symptoms and severity	11 (79%)
Prevalent among healthcare providers and in the medical system	6 (43%)
Core to BED pathology	4 (29%)
Area requiring better understanding of its trajectory and impact	4 (29%)
Traumatic ^2^	3 (21%)
Possibly varying by ethnicity ^3^	1 (7%)

**Additional participant statements regarding body weight/shape/size stigmatization and discrimination:**
*“…If the eating disorder has been associated with weight gain, then we know for a fact that they’ve been intruded upon by families, doctors … institutions, and… there’s trauma associated with that… with… being told day in and day out that what you are is not acceptable or lovable or okay,” … “many people who are living in larger bodies have been teased or bullied around weight, so there’s a lot of trauma associated binge eating disorder…” (P7)*
*“If you make a comment about somebody’s race in the middle of an airplane, as you’re getting seated …probably a bunch of people are going to [tell you] how hurtful and how unquestionably not okay that behavior is. But if you’re on that same airplane and somebody makes a comment about your weight, most people aren’t going to notice, and that’s just not okay.” (P60)*
*“Weight discrimination is legal almost everywhere in this country. … You could be fired. …based on your weight … and you have no recourse. … It’s not a protected status and it happens everywhere. …There’s research also on getting admitted to college. So that’s a similar process of applying for a job in the way people apply for college but it’s about access to higher education and there’s research showing weight discrimination comes in there.” (P16)*

Results expressed as *n* (%), in which percentages are *n*/14 times 100. ^1^ E.g., males, specific ethnicities. ^2^ Especially when occurring during childhood or chronically. ^1,3^ Suggests different ethnicities may have different levels of acceptance around weight that impact distress frequency and treatment seeking. Abbreviations: BED, binge eating disorder.

**Table 8 ijerph-19-06243-t008:** Participant statements consistent with Theme 5, “trauma and adversity”.

**Subtheme (i) Relevant Forms of Trauma/Adversity**	**7 (50%)**
Abuse (sexual, emotional, or physical)	4 (29%)
Early childhood abuse	2 (14%)
Body weight/shape/size stigmatization	3 (21%)
COVID-19 pandemic	3 (21%)
Invalidating/oppressive experiences/environments	2 (14%)
Interpersonal trauma	2 (14%)
Mandated movement or physical activity1	2 (14%)
Childhood of food scarcity/insecurity as ACES	1 (7%)
Chronic dieting	1 (7%)
Untreated diagnoses (e.g., ADHD)	1 (7%)
Impacts of IBS	1 (7%)
Trauma related to self-neglect and negative views on self-care2	1 (7%)

**Subtheme (ii) Relationship between trauma/adversity & BED**	**11 (79%)**
Trauma/adversity as relevant to BED psychopathology	11 (79%)
Trauma/adversity highly relevant for a minority with that comorbidity	1 (7%)
Trauma/adversity as increasing risk for BED	5 (36)
Cited research findings	2 (14%)
ACES can result in PTSD and BED	2 (14%)
Trauma/adversity increase risk for m[any] psychiatric problems	2 (14%)
Trauma/adversity often precede BED (not vice versa)	1 (7%)
Childhood (but not adult) trauma/adversity as risk factor	1 (7%)
PTSD highly comorbid with BED and food addiction	1 (7%)
Neurobiological impacts of trauma/adversity may prime BED	2 (14%)
Negative impact on self-regulation	1 (7%)
Binge eating to cope with trauma/adversity3	2 (14%)
Trauma/adversity as exacerbate BED symptoms	2 (14%)
Additional possible mechanistic pathways	2 (14%)
Gut microbiota as possible underlying mechanism	1 (7%)
IBS as mediator, moderator, and possible underlying mechanism	1 (7%)
Stress as possible underlying mechanism	1 (7%)
Trauma/adversity may burden BED treatment distress tolerance	1 (7%)
Trauma/adversity as comorbid/coexisting	1 (7%)

**Subtheme (iii) Critical considerations**	**5 (36%)**
Importance of addressing trauma and adversity in treatment	4 (29%)
Importance of establishing how to address trauma/adversity history in treatment	1 (7%)
Importance of screening for trauma and adversity	2 (14%)
Need for greater understanding of the relationship between trauma/adversity and BED	1 (7%)
Literature findings on poor self-report of trauma	1 (7%)

**Additional participant comments on trauma/adversity and BED**	
*“Trauma is so bad for the brain and what we’re seeing around ‘little t trauma,’ if you are someone [who is] susceptible, and you are teased and bullied, I think there’s a lifelong consequence for a lot of those individuals, and I think that absolutely sets up the trajectory around eating dysregulation, no question about it.” (P72)* *“We definitely see the association of binge eating and food addiction with PTSD [in the literature]; they are do appear to be more comorbid with each other.” (P19)* *“We know for so many people [who have childhood trauma or adversity], they’re just repeatedly re-traumatized and experience …many ACES, and then many traumas, so it just keeps building on each other. [The] childhood period is critical.” (P19)* *“I don’t think we have data to prove this is the case. But I think one of the things that having PTSD may do is it may make it harder to tolerate the distress that comes with doing treatment.” (P5)* *“[Early life trauma or adverse childhood experiences are almost] a risk factor for any psychiatric problem.” (P53)*	

**Additional comments on critical considerations regarding trauma/adversity & BED**	
*“[Trauma, adversity, and PTSD] increase …risk [for binge eating disorder] and [are] important thing[s] to ask people about to assess and to take into account when …doing formulation and recommending therapies or treatments… [When thinking about] how to help people [it is important] to have that knowledge. …There is a tendency to move towards a typical trauma-informed therapy. … [I] think …we should be taking account of people’s narrative and life experiences and that should be informing our therapy and our therapeutic approaches.” (P93)* *“I think we really need to be asking people … particularly about adverse childhood experiences” (P60)* *“[There are] a couple of great papers from …John Jay College [of] Criminal Justice in New York [that review] people’s reports of early life trauma, and … [found that] they’re highly suspect. …there are several studies where they’ve got data about kids interviewed early in life and then they interviewed them, you know, 20 30, 40 years later, and asked them about their early life experiences, and my take from [the] review, is that people’s memories are bad, that people report early life experiences as negative early life experiences more related to their parent state than to what actually happened. …It makes me worry a lot about retrospective reports of childhood experiences. I think there’s real reason to doubt their veracity.” (P46)*	

Results expressed as *n* (%), in which percentages are *n*/14 times 100. ^1^ E.g., being forced to run in gym class and ridiculed by peers. ^2^ “There’s that trauma of [the belief that] ‘taking care of myself [is] bad and selfish, and I shouldn’t do that.,’ and even if they can’t verbalize that [view], it’s there,” (P37). ^3^ Or resulting mood regulation disturbances. Abbreviations: ACES, adverse childhood experiences; ADD, attention deficit disorder; ADHD, attention deficit hyperactive disorder; ALA, adverse life experiences; BED, binge eating disorder; COVID-19, Coronavirus-19; ELA, early life adversity; IBS, irritable bowel syndrome; PTSD, post-traumatic stress disorder.

**Table 9 ijerph-19-06243-t009:** Participant statements relating to Theme 6, “interpersonal factors”.

**(i) Ways interpersonal deficits or negative interpersonal relationships can impact BED**	**7 (50%)**
Social sensitivity related to social anxiety/fear/threat perception	5 (29%)
Interpersonal deficits ^1^ affecting relationships and social support	3 (21%)
Socializing around food/eating as a problematic social activity	3 (21%)
Social anxiety as a relevant comorbidity in BED	3 (21%)
Referenced research on the role of social threat	1 (7%)
Suggested social anxiety disorder is “the most common additional mental health problem for people with an eating disorder”	1 (7%)
Spousal relationships, intimacy, and sexuality	2 (14%)
Negative social experiences or deficiency communication directly catalyzing or contributing to binge eating behavior	2 (14%)

**(ii) Ways aspects of BED can contribute to interpersonal deficits**	**5 (36%)**
Body weight/shape/size stigmatization	2 (14%)
Body weight/shape/size overvaluation	1 (7%)
Social ranking	1 (7%)
Broader social phenomenon ^2^	1 (7%)
COVID-19 quarantine/isolation	1 (7%)

**(iii) Impacts of interpersonal factors on BED pathology**	**9 (64%)** ^4^
Negative relationship between interpersonal factors and BED pathology	7 (36%)
Positive relationships between social interaction and BED pathology ^3^	3 (21%) ^4^
Positive impacts of community	2 (14%)
Benefits of family	1 (7%)
Neutral, nonspecific statement	1 (7%)

**Additional participant statements relating interpersonal factors to BED pathology:**
*“In the context of binge eating disorder, where a lot of the actual binges occur alone and in an isolated kind of way … negative interpersonal experiences can set people up to be prone to use. …the ability to interact socially with others, and the social messages we get about food and our bodies, and how important it is to fit in a certain body category also plays a role.” (P19)* *“For some people, it’s highly communicative to say, ‘I’m not telling my husband I’m mad at him but if I eat the whole pie, it’s a way of getting that across to him,’.” (P72)*
*“Certainly, folks who …have increased concerns about shape, weight, and their own appearance and body image are going to likely have more challenges socially.” (P46)* *“I think …something that is so fraught and deserves attention [is]: as people’s eating [disorders progress], how is that on [a] relationship? How is that physically? … How do they experience that from their own experience of their sexuality, how do their potential partners or current partners experience that? I think that’s an enormous impact.” (P60)*

Results expressed as *n* (%), in which percentages are *n*/14 times 100. ^1^ E.g., poor communication skills or social interaction abilities. ^2^ E.g., systemic discrimination and stigmatization. ^3^ Outside of social media and social messaging. ^4^ Two participants made statements about negative relationships between interpersonal factors and BED pathology AND about positive relationships between social interaction and BED pathology. Abbreviations: BED, binge eating disorder; COVID-19, Coronavirus-19; IP, interpersonal (relationships, skills, or deficiencies).

**Table 10 ijerph-19-06243-t010:** Participant statements relating to Theme 7, “social messaging and social media”.

**Social Messaging and Social Media**	**7 (50%)**
Significantly relevant to binge eating disorder pathology	7 (50%)
Social media as relevant	5 (36%)
Social messages as relevant	3 (21%)
Relationship described as exclusively negative ^1^	4 (28%)
Relationship described as primarily negative but with some positive aspects or potential	3 (21%)
Relationship described as exclusively positive	0 (0%)

**Additional participant statements regarding social messaging and social media:**	
** *“Social media clearly does not replace [community]; it seems to lead to more comparisons and more loneliness.” (P72)* ** *“When Meghan Trainor came out with ‘All about that base,’ …people around me were like, ‘that’s not going to make a difference,’ [but] I was like, ‘no, no, something’s shifting,’.” (P72)*

Results expressed as *n* (%), in which percentages are *n*/14 times 100. ^1^ Primarily by reinforcing ideals around body weight/shape/size, food, eating, and fitness that contribute to social ranking, social interactions, and self-esteem/valuation/negative affect.

**Table 11 ijerph-19-06243-t011:** Participant responses pertaining to Theme 8, “predatory food industry practices”.

**Subthemes Related to Food Industry Practices**	**4 (29%)**
(i) “Predatory” food industry practices described ^1^	4 (29%)
Describe foods intentionally designed to produce specific reward responses that promote excessive consumption	2 (14%)
Comparisons made between “big tobacco” and “big food” industries	2 (14%)
(ii) Food industry practices described as public policy issue	2 (14%)
(iii) Call for public education ^2^	2 (14%)
(iv) Express view that disordered eating behavior can be associated with specific foods, but can be extinguished	2 (14%)
(v) Rewarding food properties acknowledged but not described as intentionally engineered	1 (7%)

**Additional statements describing “predatory” food industry practices and environments**
*“The evidence suggests that it’s almost all ultra-processed, highly rewarding, potentially addictive foods that people are bingeing on, and in a really extreme manner and we kind of don’t acknowledge that those foods potentially have this differential impact on our reward systems, inhibitory systems, emotion regulation systems.” (P19)* *“We have … whole … city areas that are geared up around fast food. … [there are places where you can] very easily get a hold of fast food. You [can’t] very easily get a hold of decent food.” (P84)*
*Additional statements describing food/eating extinction/normalization*
*“What I’m worried about this notion of getting into it somehow the food’s fault, or it’s …an illness process that resides within the individual that’s partly exhausting to make people get well. I’ve seen too many people do too well in binge eating disorder, to believe that there’s some kind of genetic element or sometimes some kind of cue learning event going on that is unchangeable. Certainly, you get cue learning, but it’s something we actually deal with all the time, then it seems to go away if you unlearn it, which is you basically oblivion conditioning stuff.” (P84)*

Results expressed as *n* (%), in which percentages are *n*/14 times 100. ^1^ E.g., hiring engineers to design foods that produce specific rewarding or emotional responses and promote consumption, potentially leading to over-consumption and binge eating. ^2^ E.g., informing individuals with binge eating disorder of the nature of “hyper-engineered foods” and food industry practices to provide a full picture of “[the foods and industries] they’re dealing with”, (P16).

**Table 12 ijerph-19-06243-t012:** Participant statements related to theme 9, “research gaps and future directives”.

**Subtheme (i) Need for CHANGE in the systems That Abet BED**	**10 (71%)**
(a) Eating Disorder Field	5 (36%)
Eating disorder research funding ^1^	2 (14%)
Mandated movement perpetuated by healthcare system	2 (14%)
Recognizing implicit weight bias/stigma/discriminating in the field	1 (7%)
(b) Food systems & availability	4 (29%)
Food industry practices	2 (14%)
Food stamp allotment	1 (7%)
(c) Other systems of oppression ^2^	2 (14%)
(d) Economic aspects that prevent treatment access	1 (7%)

**Subtheme (ii) Understanding the role of environmental impact/risk factors on BED**	**5 (36%)**
(a) Traumatic impacts of mandated movement	2 (14%)
(b) Impacts of trauma	1 (7%)
(c) Impacts of “broader sociocultural issues”	1 (7%)
(d) Impacts of community	1 (7%)
(e) Impacts of interpersonal threat/threat sensitivity	1 (7%)
(f) Impact of environmental pollution	1 (7%)

**Subtheme (iii) Inclusion of minority and marginalized populations**	**4 (29%)**
(a) Including and reaching men	1 (7%)
(b) Including individuals in normal-sized bodies	1 (7%)
(c) Identifying struggles unique to marginalized populations	1 (7%)
(d) Information dissemination ^3^	1 (7%)

**Subtheme (iv) Recognizing and understanding weight bias/stigma/discrimination**	**4 (29%)**
(a) Research investigation of forms, prevalence, and impacts	4 (29%)
(b) Recognizing implicit weight bias/stigma/discrimination in the field	1 (7%)

**Subtheme (v) Taking & understanding the narrative of individuals with BED**	**3 (21%)**
(a) Identifying how to “*listen for what people are telling us about their experience*?”	1 (7%)
(b) Listening to- and understanding the unique experiences of individuals with BED	1 (7%)

**Subtheme (vi) Understanding consequences of BED**	**2 (14%)**
(a) Impacts on interpersonal relationships	2 (14%)
(b) Impacts on threat sensitivity	1 (7%)
(c) Impacts on expression of sexuality	1 (7%)

**Participant statements conveying the need for change in the systems that abet BED**	
*“Thinking about … the larger systemic issues that might set people up for some of this. …there’s been some research on **food insecurity** and eating disorders, particularly as it pertains to binge eating. … I work with patients who have said ‘well yeah I have binge eating. I binge eat the first two weeks of the month ‘cause that’s when we have food in the house and then there’s no food in the house the last two weeks of the month.’ **That’s a systemic issue that I think needs to be addressed and needs to be talked about in terms of people’s vulnerability to eating disorders**.” (P75)*	

Results expressed as *n* (%), in which percentages are *n*/14 times 100. ^1^ Including need for more funding (equally proportionate to that available for research on other disorders of similar magnitude) and need for clarification on which funding agencies should fund eating disorder research. ^2^ E.g., structural racism and sexism, economic exploitation (see statements from P16 in section A), and “broader sociocultural issues”. ^3^ “What do we do then to reach these [marginalized] communities in a way that’s meaningful?” Abbreviations: BED, binge eating disorder.

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
