# Peer review of "Binge Eating Disorder Is a Social Justice Issue: A Cross-Sectional Mixed-Methods Study of Binge Eating Disorder Experts’ Opinions"

_ijerph, 2022, doi:10.3390/ijerph19106243_

Round 1
Reviewer 1 Report
Dear authors,
thank you for interesting study!
My suggestions and recommendations about the article:
- It is necessary to add the topicality of the research to the introduction.
- The introduction should be supplemented by research done so far on the current topic.
- The second paragraph of the introduction (lines 41-49) should be moved to the section: methods.
- The introduction does not state the aim of the study. It is not entirely clear what the authors wanted to prove with this study.
- Line 47 has incorrect numbering, number 3 should be followed by number 4 instead of 5. Total 9 themes instead of 10 (line 44).
- Section 3.0. Participant response rates and characteristics with the Figure 1 and the Table 3 is the methodology, not the results.
- The information provided on lines 103-106 is repeated. See lines 89-94.
Author Response
Reviewer 1:
My suggestions and recommendations about the article:
- It is necessary to add the topicality of the research to the introduction. Thank you for this invaluable feedback. We have added background information to the introduction section (lines 41-53) that we hope will add to the topicality of the research and clarify that binge eating disorder is a field in which empirical findings on the contributions of environmental factors is lacking.
- The introduction should be supplemented by research done so far on the current topic. Thanks. As per above, we have added background information to the introduction section that provides a background to the research that has been done so far on the current topic.
- The second paragraph of the introduction (lines 41-49) should be moved to the section: methods. Thank you. When we revised the introduction, this second paragraph was removed.
- The introduction does not state the aim of the study. It is not entirely clear what the authors wanted to prove with this study. Thank you. We have added the following sentence to the end of the introduction (lines 57-58): “Therefore, the aim of this study was to collect information on environmental factors that experts in the field deem relevant to adult binge eating disorder pathology.”
- Line 47 has incorrect numbering, number 3 should be followed by number 4 instead of 5. Total 9 themes instead of 10 (line 44). Thanks. Per our response to #3, this paragraph (and the incorrect numbering it contained) has been removed.
- Section 3.0. Participant response rates and characteristics with the Figure 1 and the Table 3 is the methodology, not the results. Thank you. We have moved section 3.0 to section 2.4 in the methods, along with Fig 1 and Table 3, as you have suggested.
- The information provided on lines 103-106 is repeated. See lines 89-94. Unfortunately, we were not able to identify this repeated information based on the line numbers you provided (we suspect the line numbers on our draft do not match the one on yours). We sincerely apologize for this inconvenience. If you would be so kind as to copy/paste the repetitive phrases, we will be happy to accommodate this change.
- We had another oversight that we caught in review, and accordingly, we added the following paragraph to the bottom of section 4.2:
“Another important limitation of this study is that demographic data was collected for sex assigned at birth, but not for gender. This is a shortcoming for two reasons: 1) gender is relevant to this study question and sex assigned at birth is not, and 2) asking for sex assigned at birth follows an old convention (collecting information on sex assigned at birth by default rather than collecting only the most relevant information for the study question) that fails to include and account for equity and diversity. While the question of sex assigned at birth is not relevant to this study question, the question of gender is. It would also have been relevant and important to identify whether any study participants identify as agender or transgender since binge eating disorder has higher prevalence among transgender and gender non-binary individuals (at least in youth and young adults; to the authors’ knowledge we are still lacking this information in adults) [25, 37]. Overall, the field should work toward developing a base of researchers and clinicians who study and treat binge eating disorder that more accurately represents those who experience it.”
Reviewer 2 Report
This cross-sectional study looked at the impact of environmental factors on adult binge eating disorder pathology as perceived by experts in the field. Please address the issues listed below.
- The abstract did not include any findings or conclusions.
- The ten themes identified in the current study must clearly introduce the background(s).
- The backgrounds of 14 experts were not presented in detail.
- The experts interviewed expressed a growing awareness of systemic issues associated with binge eating disorder. However, it appears to be limited in the current study's questions.
- The experts were not invited from countries other than the United States, most likely due to NIH funding criteria. How can we avoid this bias in the discussion?
- The current study's novelty was not clearly demonstrated.
Author Response
Reviewer 2:
This cross-sectional study looked at the impact of environmental factors on adult binge eating disorder pathology as perceived by experts in the field. Please address the issues listed below.
- The abstract did not include any findings or conclusions. We thank you for this feedback. We have separated the abstract into the segments of Background, Methods, Results, and Conclusions.
- The ten themes identified in the current study must clearly introduce the background(s). Thank you. We have added information to the background section.
- The backgrounds of 14 experts were not presented in detail. We thank you for this feedback. The experts are anonymous, so we cannot present more information than that provided in Tale 1 (“Participant eligibility criteria and number of participants who were recruited, and who consented and enrolled,”) and Table 3 (“Characteristics of study participants”).
- The experts interviewed expressed a growing awareness of systemic issues associated with binge eating disorder. However, it appears to be limited in the current study's questions. Thank you. The findings conveyed in this manuscript come from a larger study in which the same 14 experts were interviewed about their perspectives on adult binge eating disorder pathology and treatment broadly. Because of the robust commentary regarding binge eating disorder as a social justice issue, the authors thought the theme was substantive enough for a separate manuscript, even if not the exclusive focus of the interviews. Perhaps if the interviews had focused solely on environmental factors relevant to adult binge eating disorder pathology, perhaps we could have delved further into the social justice issues deemed relevant to adult binge eating disorder. To clarify this – as recognize the limitation you identified, we added the following paragraph to the end of the limitations section:
“Additionally, it should be noted the findings reported here come from a larger study that collected information from the fourteen experts reported here on their perspectives of adult binge eating disorder pathology and treatment. Because of the robust commentary regarding binge eating disorder as a social justice issue, the authors felt the theme was substantive enough for a separate manuscript, even if not the exclusive focus of the interviews. If the interviews had focused solely on environmental factors relevant to adult binge eating disorder pathology, perhaps we could have delved further into the social justice issues deemed relevant to adult binge eating disorder.”
5. The experts were not invited from countries other than the United States, most likely due to NIH funding criteria. How can we avoid this bias in the discussion? Although NIH R01, T32, and P grant funding was 1 of 4 possible eligibility criteria for researchers (Table 1, section I), this did not preclude researchers from other countries to participate. In fact, our study included researchers from the UK, AU, and CA, as well from the US (see “Geographical location of residence” section in Table 3). The bias toward researchers in the U.S. is acknowledged at the end of the discussion (last sentence in the 2nd paragraph of section 4.3, “Study Limitations & Strengths,”). To better acknowledge the limitation of this bias, we have also added lines 928-936 to the first paragraph of section 4.3, which state:
“Moreover, NIH R01, T32, or P grant funding was one of four possible eligibility criteria that researchers were required to meet for participation eligibility (Table 3). Eligible participants were required to meet one of the four criteria presented in section I of Table 3 (not all four). Thus, our study did not limit researcher participation to U.S. researchers. However, this criterion did present a bias for inclusion of academics and researchers within the U.S. Although the study sample does include individuals from the UK, AU, and CA, 50% of study participants were American. Thus, the themes identified here must be interpreted with caution when done in the context of binge eating disorder pathology globally (vs. within the U.S.)..”
6. The current study's novelty was not clearly demonstrated. Thank you for this feedback! We have restructured the “Study Strengths” section (4.3) to emphasize the most novel aspect of the study (identified in the first 2 sentences, stating:
“The study of social justice issues relative to their influences on behaviors is a relatively new contribution to “hard”science. This movement provides a new lens through which to view eating disorders. Our study aims in broaden the awareness of social justice issues that are both important and historically overlooked in the field of adult binge eating disorder. To the authors’ knowledge, this study is the first to synthesize expert opinion on environmental factors contributing to adult binge eating disorder pathology.”
- We had another oversight that we caught in review, and accordingly, we added the following paragraph to the bottom of section 4.2:
“Another important limitation of this study is that demographic data was collected for sex assigned at birth, but not for gender. This is a shortcoming for two reasons: 1) gender is relevant to this study question and sex assigned at birth is not, and 2) asking for sex assigned at birth follows an old convention (collecting information on sex assigned at birth by default rather than collecting only the most relevant information for the study question) that fails to include and account for equity and diversity. While the question of sex assigned at birth is not relevant to this study question, the question of gender is. It would also have been relevant and important to identify whether any study participants identify as agender or transgender since binge eating disorder has higher prevalence among transgender and gender non-binary individuals (at least in youth and young adults; to the authors’ knowledge we are still lacking this information in adults) [25, 37]. Overall, the field should work toward developing a base of researchers and clinicians who study and treat binge eating disorder that more accurately represents those who experience it.”